# Women with polycystic ovary syndrome exhibit impaired endometrial receptivity with excessive ERα and histone lactylation

Hongying Shan [1,2,3,4,5,7], Yue Wang [1,2,3,4,7], Baoying Liao[1,2,3,4,7],
Kai-Lun Hu[1,2,3,4,7], Xiunan Chen[1,2,3,4], Chenxi Xiao[1,2,3,4], Zi Yang[1,2,3,4],
Fenting Liu[1,2,3,4], Tianliu Peng[1,2,3,4], Mingmei Lin[1,2,3,4], Feng Deng[1,2,3,4],
Ping Zhou[1,2,3,4] ✉, Yang Yu [1,2,3,4,6] ✉, Rong Li [1,2,3,4] ✉ & Heng Pan [1,2,3,4] ✉

Polycystic ovary syndrome (PCOS) is one of the most common reproductive disorders in women and severely impairs fertility. Extant clinical studies can only provide indirect and plausible evidence to support endometrial dysfunction as an ovary-independent contributor to PCOS infertility, considering heterogeneous confounders in their phenotypes, comorbidities, and severities. By strictly controlling embryonic factors and potential confounders, our retrospective cohort study reports an adverse implantation rate in women with PCOS, confirming abnormalities in the endometrium, which are accompanied by excessive ERα and histone lactylation. Next, we validate the cooccurrence of impaired uterine receptivity with elevated ERα and histone lactylation in the PCOS mouse model. Inhibiting histone lactylation could downregulate ERα and estrogen-responsive genes, restore uterine receptivity, and improve the implantation rate in PCOS mice. Here, we show that upregulated ERα and histone lactylation are key indicators of impaired endometrial receptivity in PCOS, providing a potential therapeutic strategy by inhibiting lactate production.

Polycystic ovary syndrome (PCOS) is the most common reproductive disorder in women of childbearing age (10–13%), characterized by irregular menstrual cycles, hyperandrogenism, and metabolic disturbance[1]. Women with PCOS have a 15-fold increased risk of infertility compared to women without PCOS[2]. Over the past decade, consensus conferences, guidelines, and position papers have stated that infertility caused by PCOS is almost exclusively due to oligo-ovulation or anovulation, and all available treatments have focused on improving ovulatory function[3]. However, accumulating evidence has raised concerns about other contributors to severe adverse pregnancy outcomes of PCOS[4,5]. For instance, the reduced fertility in ovulatory women with PCOS cannot be entirely attributed to the oocyte component[6].

In human reproduction, the embryo, the endometrium, and their interaction are essential for a successful pregnancy. Besides defects in oocytes and embryos, several clinical and biochemical factors associated with PCOS can impair endometrial function[7]. Nonetheless, whether extant clinical studies are strong enough to support the

[1]State Key Laboratory of Female Fertility Promotion, Center for Reproductive Medicine, Department of Obstetrics and Gynecology, Peking University Third Hospital, Beijing, China. [2]Key Laboratory of Assisted Reproduction, Peking University, Ministry of Education, Beijing, China. [3]Beijing Key Laboratory of Collaborative Innovation in Frontier Technologies for Population Quality, Beijing, China. [4]National Clinical Research Center for Obstetrics and Gynecology, Peking University Third Hospital, Beijing, China. [5]Reproductive Medicine Department, The First Affiliated Hospital of Shihezi University, Shihezi, Xinjiang, China. [6]Beijing Advanced Center of Cellular Homeostasis and Aging-Related Diseases, Institute of Advanced Clinical Medicine, Peking University, Beijing, China. [7]These authors contributed equally: Hongying Shan, Yue Wang, Baoying Liao, Kai-Lun Hu. ✉e-mail: zhoup0520@163.com; yuyang5012@hotmail.com; roseli001@sina.com; hep2007@bjmu.edu.cn

endometrial factor as an independent contributor to PCOS infertility remains controversial. Most studies have focused on oocyte competence and embryo aneuploidy, providing only indirect clinical evidence[6,8]. The remaining studies either haven't strictly controlled for embryonic factors and other confounders or haven't applied proper inclusion and exclusion criteria, resulting in conflicting conclusions[5,9,10].

The endometrium plays a crucial role in maintaining pregnancy and undergoes dynamic changes during the window of implantation (WOI), decidualization, and placentation. WOI, or endometrial receptivity, occurs from the 20 to 24 days of the menstrual cycle and is critical for embryo adhesion, invasion, and implantation[11]. These processes are tightly regulated by ovarian hormones (estrogen and progesterone), which affect cell proliferation, cell differentiation, and cell–cell communication via signaling pathways and transcriptional regulation[12,13]. The imbalance between the estrogen and progesterone signaling can disrupt implantation and decidualization, lead to pregnancy failure or pregnancy-related complications, and ultimately affect maternal-fetal health[14–16]. Abnormalities in estrogen/progesterone receptors (ER/PGR) are often observed in the endometrium of women with PCOS[3]. However, whether and how the imbalance affects endometrial function in PCOS remains unclear.

We aimed to connect the estrogen/progesterone signaling to epigenetic modifications for two reasons. First, appropriate epigenetic modifications, such as histone modifications, are indispensable to normal endometrial function[17,18]. For instance, H3K27ac enrichment at key genes, such as *IGFBP1*, *WNT4*, and *ZBTB16*, promotes decidualization[18,19]. Histone lactylation (histone lysine lactylation, Kla), a less studied histone modification, is also closely associated with endometrial/uterine function. In animal models, uterine lactate and Kla increase during decidualization, particularly at the implantation site (IS)[20,21]. Moreover, epigenetic modifications are frequently disrupted in endometrial aging and endometrial disorders[22,23]. Second, it has been reported that hormonal signaling pathways and epigenetic modifications can regulate each other in different tissues and diseases[23].

In this work, we hypothesize that women with PCOS are suffering from endometrial dysfunction and consequent fertility decline, which results from abnormal estrogen and/or progesterone signaling along with epigenetic alteration. To assess the endometrial impact on fertility independent of embryonic factors in PCOS, we conduct a retrospective cohort study ($n = 4278$) to evaluate pregnancy outcomes between PCOS and non-PCOS controls, after adjusting for potential confounders via propensity score matching (PSM). We further employ primary endometrial samples from patients and the PCOS mouse model to confirm that ERα overexpression, rather than abnormal PGR expression, is responsible for impaired endometrial receptivity. We next select Kla to study the interplay between ER signaling and epigenetic modifications, given that women with PCOS are reported to have elevated serum lactate levels[24]. To evaluate whether ER signaling affects Kla, we generate *ESR1*-overexpressing (ESR1-OE) Ishikawa cells and apply Methyl-piperidino-pyrazole (MPP, an ERα inhibitor) to Ishikawa cells, confirming that ERα can regulate H3K18la (a primary subtype of Kla) via LDHA/LDHB. We also manipulate lactate levels in Ishikawa cells and the uterus of PCOS mice to determine the regulatory role of Kla on ERα, demonstrating that inhibiting lactate production decreases H3K18la, mitigates ERα overexpression, and restores impaired uterine receptivity. Together, our study lays a foundation for understanding endometrial dysfunction in PCOS and highlights the potential to improve PCOS patients' fertility in the clinic by targeting Kla.

## Results

### Women with PCOS show abnormal endometrial receptivity and ERα

Our retrospective cohort study included 4278 patients, divided into PCOS and control groups (Supplementary Fig. 1). Patients in this cohort underwent high-quality single-blastocyst transfer in frozen-thawed embryo transfer (FET) cycles, aiming to strictly control embryonic factors. Potential confounders were adjusted for using PSM, resulting in favorable matching (Supplementary Fig. 2 and Supplementary Table 1). Women with PCOS demonstrated a decline in the implantation rate, positive pregnancy rate, and live birth rate

**Table 1 | Pregnancy outcomes of the two groups before and after matching in FET cycles (%)**

| Item | Before PSM | | P value | After PSM | | P value |
|---|---|---|---|---|---|---|
| FET cycles | Control group ($n = 2629$) | PCOS group ($n = 1649$) | | Control group ($n = 1406$) | PCOS group ($n = 1406$) | |
| Positive pregnancy rate (%) | 1517 (57.7) | 905 (54.9) | 0.075 | 827 (58.8) | 769 (54.7) | 0.030 |
| Chemical pregnancy rate (%) | 1473 (56.0) | 883 (53.5) | 0.120 | 801 (57.0) | 749 (53.3) | 0.053 |
| Implantation rate (%) | 1265 (48.1) | 755 (45.8) | 0.146 | 701 (49.9) | 643 (45.7) | 0.031 |
| Miscarriage rate (%) | 196 (7.5) | 161 (9.8) | 0.029 | 108 (7.7) | 136 (9.7) | 0.147 |
| Premature birth rate (%) | 89 (3.4) | 66 (4.0) | 0.573 | 54 (3.8) | 57 (4.1) | 0.854 |
| Live birth rate (%) | 987 (37.5) | 543 (32.9) | 0.008 | 551 (39.2) | 462 (32.9) | 0.002 |
| Gestational age at delivery[a] (Median [Q1, Q3]) | | | | | | |
| Singleton, weeks | 39 [38, 39] | 39 [38, 39] | 0.636 | 38 [38, 39] | 39 [38, 39] | 0.766 |
| Twin, weeks | 36 [36, 37] | 36 [35, 36] | 0.476 | 36 [36, 36] | 36 [35, 36] | 0.689 |
| Birthweight (Mean ± SD) | | | | | | |
| Singleton, g. | 3345.3 ± 496.1 | 3344.4 ± 568.9 | 0.763 | 3332.2 ± 497.5 | 3349.6 ± 553.7 | 0.506 |
| Twin, g. | 2463.3 ± 299.3 | 2422.5 ± 198.4 | 0.515 | 2425.0 ± 317.7 | 2422.5 ± 198.4 | 0.985 |

Continuous variables were tested for normality using Shapiro–Wilk test. Normally distributed data are presented as Means ± SD and compared using two-tailed unpaired Student's *t* test. Non-normally distributed data are presented as Medians [Q1, Q3] and compared using two-tailed unpaired Mann–Whitney *U* rank-sum test. Categorical variables are summarized as numbers (percentages) and compared using chi-square test. All statistical analyses were performed in R.

*PSM* propensity score matching, *FET* frozen-thawed embryo transfer, *PCOS* polycystic ovary syndrome, positive number/total number in brackets.

[a]Singleton pregnancies: the gestational ages at delivery for the control and PCOS groups were 38.32 ± 1.62 weeks and 38.18 ± 1.95 weeks before PSM, and 38.30 ± 1.68 weeks and 38.19 ± 1.90 weeks after PSM. Twin pregnancies: the gestational ages at delivery for the control and PCOS groups were 36.00 ± 1.10 weeks and 35.50 ± 1.43 weeks before PSM, and 35.80 ± 1.10 weeks and 35.50 ± 1.43 weeks after PSM.

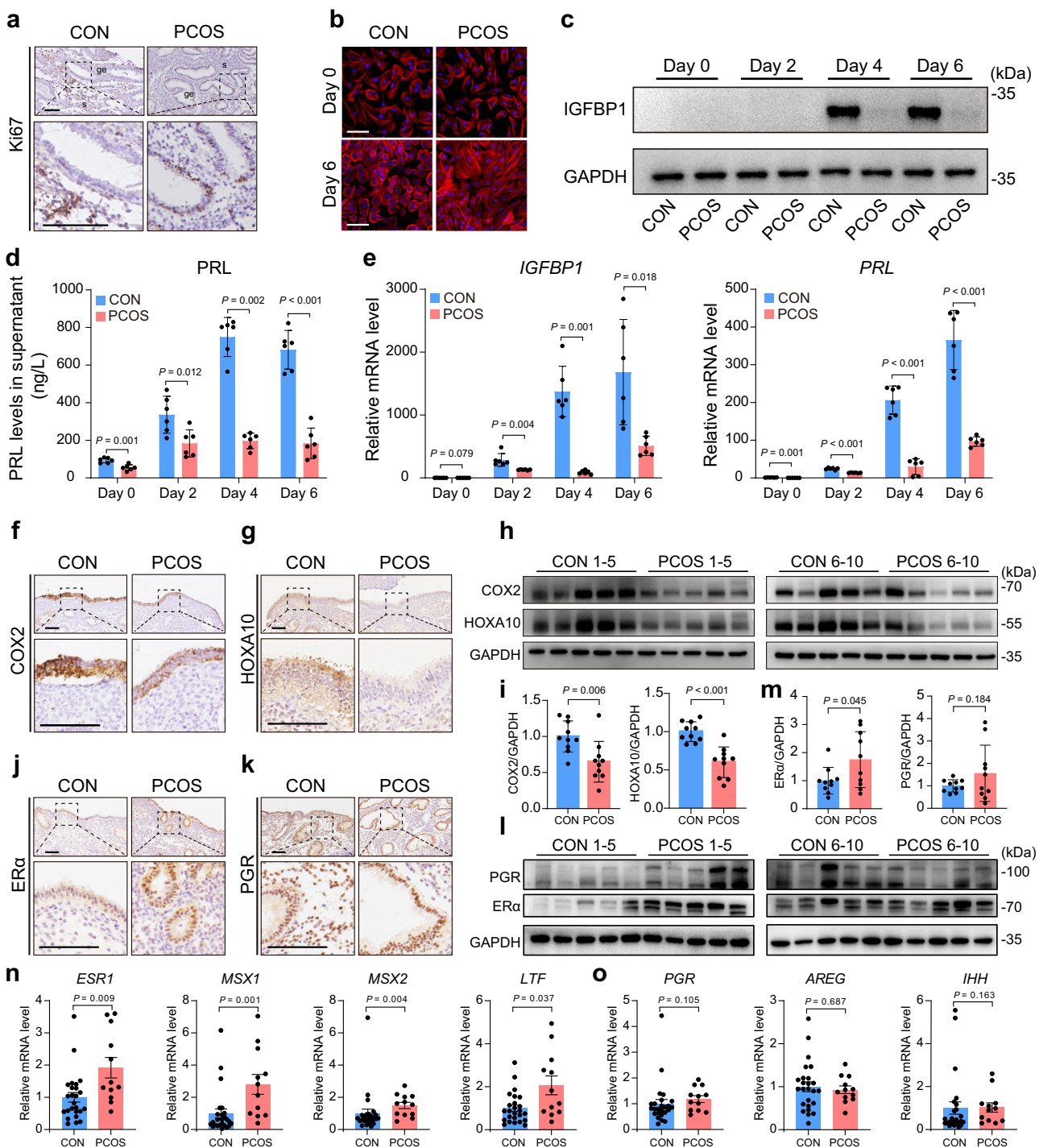

compared to controls (Table 1), suggesting that the fertility decline in PCOS is associated with dysregulated endometrial function.

The abnormal endometrium frequently exhibits impaired receptivity, characterized by extensive morphological and molecular changes[25,26]. To evaluate the expression profiling and function of endometrial cells, viable endometrial epithelial and stromal cells were successfully isolated (Supplementary Fig. 3a–f). We observed excessive epithelial proliferation and impaired stromal decidualization in PCOS, suggesting a defective mid-secretory endometrial function (Fig. 1a–e and Supplementary Fig. 3g, h). In PCOS, endometrial stromal cells failed to acquire the typical morphology of decidual cells following treatment with cAMP and medroxyprogesterone acetate (MPA) (Fig. 1b). Key decidual markers IGFBP1 and PRL were significantly

reduced in PCOS during decidualization (Fig. 1c–e and Supplementary Fig. 3h). We also observed the downregulation of COX2 and HOXA10 in the mid-secretory endometrium of PCOS, indicating impaired endometrial receptivity (Fig. 1f–i and Supplementary Fig. 3i, j). Given that estrogen and progesterone closely regulate endometrial receptivity, we subsequently assessed the expression of their receptors and responsive genes. ERα was upregulated in the mid-secretory endometrium of PCOS, whereas PGR did not change significantly (Fig. 1j–o and Supplementary Fig. 3k, l). Estrogen-responsive genes, including *MSX1, MSX2,* and *LTF*, were abnormally elevated in PCOS, whereas no significant differences were observed in progesterone-responsive genes (Fig. 1n, o). Therefore, we speculated that endometrial dysfunction in PCOS is likely associated with excessive ERα expression.

**Fig. 1 | PCOS endometrium has abnormal endometrial receptivity and ERα expression. a** Ki67 immunohistochemistry (IHC) staining in the human mid-secretory endometrium. ge glandular epithelium, s stroma. **b** Immunofluorescence (IF) staining of cell morphologies in human endometrial stromal cells during induced decidualization. Decidualization was induced by the treatment with 0.5 μM cAMP and 1 μM MPA. The F-actin cytoskeleton was visualized by rhodamine phalloidin staining. **c** Representative images illustrating protein levels of IGFBP1 in human proliferative endometrial stromal cells during induced decidualization (n = 5, biologically independent samples). **d** Protein levels of PRL in cell culture supernatants of human proliferative endometrial stromal cells during induced decidualization (n = 6, biologically independent samples). **e** Relative mRNA levels of *IGFBP1* and *PRL* in human proliferative endometrial stromal cells during induced decidualization (n = 6, biologically independent samples). **f**, **g** COX2 and HOXA10 IHC staining in the human mid-secretory endometrium. **h**, **i** Protein levels of COX2 and HOXA10 in the human mid-secretory endometrium (n = 10, biologically

independent samples). **j**, **k** ERα and PGR IHC staining in the human mid-secretory endometrium. **l**, **m** Protein levels of ERα and PGR in the human mid-secretory endometrium (n = 10, biologically independent samples). **n**, **o** Relative mRNA levels of *ESR1*, *MSX1*, *MSX2*, *LTF*, *PGR*, *AREG*, and *IHH* in the human mid-secretory endometrium (n = 25 for CON and n = 12 for PCOS, biologically independent samples). Exact P values: **d** day 6, $P = 3.15 \times 10^{-6}$; **e** Relative mRNA levels of *PRL* on day 2, $P = 1.43 \times 10^{-4}$; day 4, $P = 1.42 \times 10^{-6}$; day 6, $P = 3.26 \times 10^{-4}$; **i** Protein levels of HOXA10, $P = 4.78 \times 10^{-5}$. For **d**, **e**, **i**, **m**, relative mRNA levels of *LTF* in (**n**) and relative mRNA levels of *AREG* in (**o**), P values were determined by two-tailed unpaired Student's t test, and data are presented as means ± s.e.m. For relative mRNA levels of *ESR1*, *MSX1*, and *MSX2* in (**n**) and relative mRNA levels of *PGR* and *IHH* in (**o**), P values were determined by two-tailed unpaired Mann–Whitney U rank-sum test, and data are presented as medians with interquartile ranges. For **a**, **b**, **f**, **g**, **j**, and **k**, images are representative of four independent biological replicates. Scale bar: 100 μm. Source data are provided as a Source data file.

## Women with PCOS exhibit elevated endometrial Kla

Lactate-derived Kla is crucial for embryo implantation and decidualization[20,27]. We first assessed Pan Kla in the human endometrium. Pan Kla level was minimal in the proliferative endometrium, while increased markedly in the secretory stage and the decidua (Supplementary Fig. 4a), which was consistent in the murine uterus (Supplementary Fig. 4b). As the two most important types of Kla[28–30], H3K18la and H4K12la both showed upregulation in the secretory endometrium and decidua (Fig. 2a and Supplementary Fig. 4c–f). These observations indicated that the increased Kla synchronizes with endometrial receptivity.

We next investigated whether the Kla dynamic was disturbed in PCOS. PCOS secretory endometrium exhibited a higher intracellular lactate level (Fig. 2b), consistent with the elevated serum lactate level in PCOS[24]. Pan Kla, H3K18la, and H4K12la levels were significantly upregulated in the PCOS mid-secretory endometrium and in PCOS patient-derived organoids compared with the control (Fig. 2c–i and Supplementary Fig. 4g–j). To further understand the regulatory impact of Kla, we aimed to characterize local changes between the PCOS and control groups. Compared with H4K12la, H3K18la has been extensively studied[30,31]. Also, the H3K18la ChIP antibody is more stable and widely used[32,33]. Therefore, H3K18la CUT&Tag was applied to the mid-secretory endometrium of PCOS and controls, demonstrating high consistency across biological replicates (Fig. 2j and Supplementary Fig. 4k). Local H3K18la levels were also elevated in PCOS, especially at promoters (Fig. 2j and Supplementary Fig. 4l). Genes with additional H3K18la signals in PCOS were enriched in pathways related to steroid hormone (*ESR1*, *MSX1-2*, and *LTF*) and cell cycle (*ERBB2-4* and *CCND1*) (Fig. 2k–m)[34–36], consistent with observed abnormal hormone response and epithelial proliferation, which are associated with impaired endometrial receptivity.

## ERα and Kla upregulate each other upon their increase

To understand the regulatory relationship between ERα and Kla, we first generated *ESR1*-overexpressing (ESR1-OE) Ishikawa cells via lentiviral transduction and confirmed their normal cell viability (Supplementary Fig. 5a–c). We observed increased levels of both ERα and H3K18la (Fig. 3a–c and Supplementary Fig. 5d–f), confirming the successful overexpression of *ESR1* and its impact on H3K18la. Our next question is how ERα overexpression promotes the increase of H3K18la. Kla is primarily induced by intracellular lactate accumulation[37]. Besides H3K18la, the intracellular lactate level was also increased in ESR1-OE cells (Fig. 3d). LDHA and LDHB are two major enzymes involved in lactate production[38,39], which were increased upon *ESR1* overexpression and are potentially responsible for the elevated intracellular lactate (Fig. 3e–g). We also assessed the gene expression of Kla writers and erasers. No significant changes in *EP300* and *SIRT1-3* were observed between ESR1-OE cells and controls (Fig. 3h), indicating that accumulated intracellular lactate levels, rather than changes in Kla

regulators, induce H3K18la increase upon *ESR1* overexpression. Lastly, we explored whether manipulating Kla levels could affect ERα. Sodium lactate (Nala) was added to Ishikawa cells to increase the intracellular lactate level. A 25 or 50 mM Nala treatment increased H3K18la and ERα in Ishikawa cells (Supplementary Fig. 5g–i). As an active transcription mark[40], we observed an H3K18la signal at the *ESR1* promoter and speculated that the increased H3K18la level can elevate *ESR1* expression (Fig. 2m).

## PCOS mice have abnormal uterine receptivity, ERα, and H3K18la

We further validated our observations using a DHEA-induced PCOS mouse model[41]. Compared to controls, the DHEA group showed PCOS-specific reproductive phenotypes, including disrupted estrous cycles (Fig. 4a, b and Supplementary Fig. 6a), increased cystic follicles (Fig. 4c), reduced corpora lutea (Fig. 4c), as well as increased levels of testosterone, luteinizing hormone, and estradiol (Supplementary Fig. 6b). For metabolic phenotypes, we observed elevated body weights, glucose intolerance, and insulin resistance in the DHEA group (Supplementary Fig. 6c–h). These observations confirmed the successful establishment of the PCOS mouse model. After mating with normal fertile male mice, the DHEA group had no ISs on day 5, even though many unimplanted high-quality blastocysts could be identified upon flushing uterine horns (Fig. 4d, e). To further exclude embryonic factors, normal blastocysts from control mice were transferred into the uteri of the DHEA group on day 4. Similarly, no ISs were observed on day 5, with detectable unimplanted blastocysts flushed from uterine horns (Fig. 4f, g), confirming the abnormal uterus in the DHEA group. We next assessed uterine receptivity in the DHEA and control groups. Consistent with PCOS patients, DHEA mice exhibited phenotypes of impaired uterine receptivity, including excessive branching structures, epithelial proliferation, and stromal cell apoptosis (Fig. 4h–j and Supplementary Fig. 6i–k). ERα, not PGR, was upregulated in the uterus of the DHEA group compared with controls (Fig. 4k–n and Supplementary Fig. 6l, m). Uterine H3K18la was also significantly higher in the DHEA group (Fig. 4o–q). Collectively, the cooccurrence of impaired endometrial/uterine receptivity, ERα overexpression, and H3K18la increase in PCOS was conserved between humans and mice.

## Reducing ERα/Kla recovers uterine receptivity in PCOS mice

To determine the causal relationship between changes in ERα/Kla and uterine receptivity, we reduced ERα or Kla in the DHEA mice. First, we administered the ERα inhibitor MPP into the uterine horns of DHEA-treated mice on day 3 (Supplementary Fig. 7a). By day 5, MPP-injected DHEA mice exhibited significantly increased ISs compared to the DHEA group (Supplementary Fig. 7a–c), featuring restored uterine branching and lower H3K18la levels (Supplementary Fig. 7d–f). Second, we injected a glycolysis inhibitor, oxamate, into the uterine horns of DHEA mice on day 3 (Fig. 5a). There was a significant increase in both the number of ISs and the percentage of mice with at least one IS on day 5

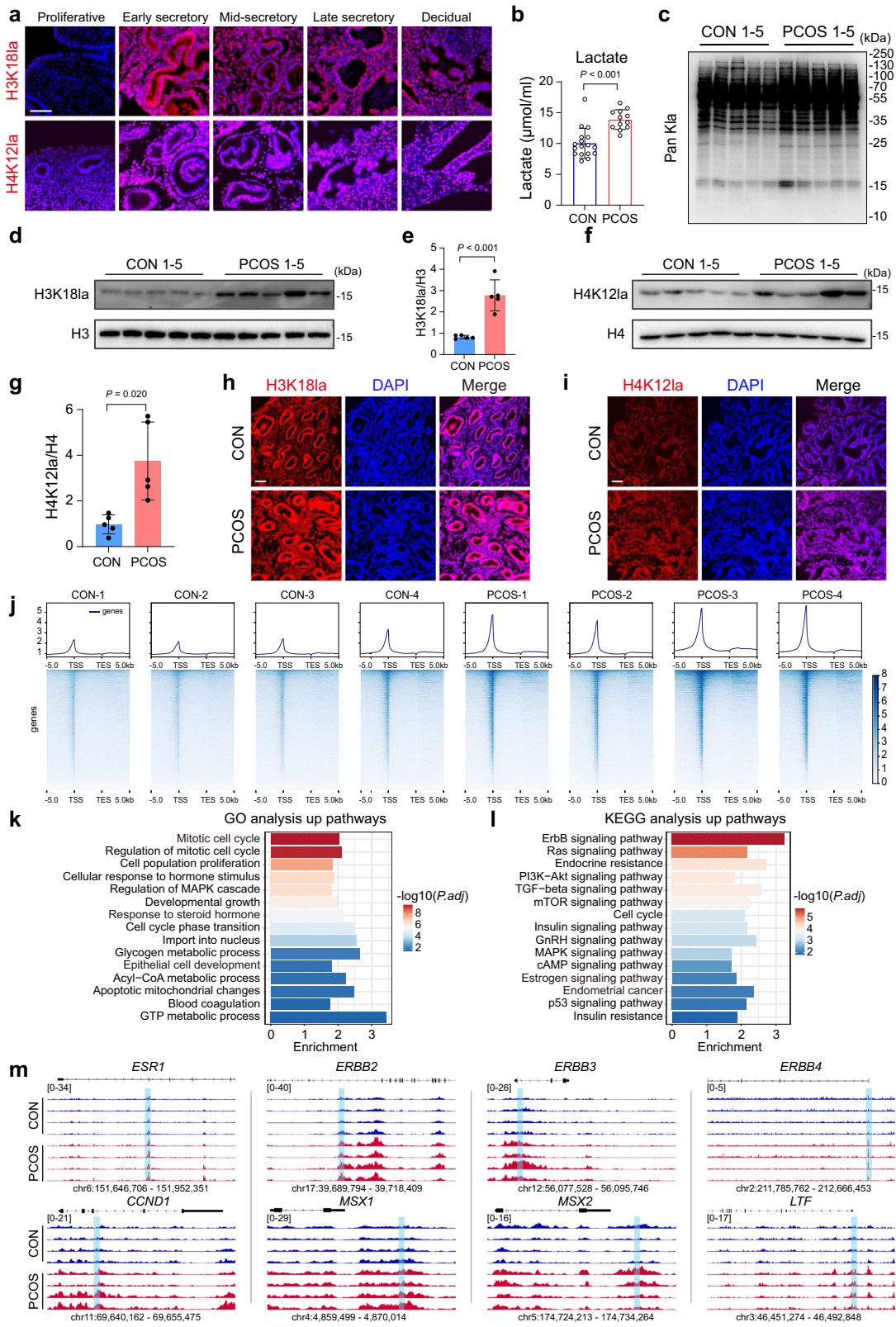

in DHEA mice injected with the 2.25 mg/kg oxamate compared to the DHEA group (Fig. 5b–d). Uterine lactate and H3K18la levels were also decreased upon the oxamate treatment (Fig. 5e–g and Supplementary Fig. 7g), along with recovered uterine status (Fig. 5h and Supplementary Fig. 7h, i). The abnormalities in ERα and estrogen-responsive genes were also partially eliminated by oxamate (Fig. 5i–k and Supplementary Fig. 7j). These findings suggest that ERα or Kla reduction can rescue the abnormal uterine receptivity and implantation failure, improving the fertility of PCOS mice.

## Discussion

PCOS is a common endocrine disorder, and growing evidence suggests that PCOS negatively impacts fertility and pregnancy outcomes[5]. Besides defective qualities of oocytes and embryos, endometrial

**Fig. 2 | PCOS endometrium shows elevated Kla. a** H3K18la and H4K12la IF staining in the human endometrium of different menstrual stages. **b** Intracellular lactate levels in the human mid-secretory endometrium ($n = 17$ for CON and $n = 12$ for PCOS, biologically independent samples). **c** Pan Kla levels in the human mid-secretory endometrium ($n = 5$, biologically independent samples). **d–g** H3K18la and H4K12la levels in the human mid-secretory endometrium ($n = 5$, biologically independent samples). **h, i** H3K18la and H4K12la IF staining in the human mid-secretory endometrium. **j** The H3K18la signal around gene bodies in the human mid-secretory endometrium ($n = 4$, biologically independent samples). **k** GO enrichment analysis of genes with H3K18la increase in the PCOS mid-secretory endometrium. **l** KEGG enrichment analysis of genes with H3K18la increase in the PCOS mid-secretory endometrium. **m** The H3K18la signal in the human mid-secretory endometrium at selected genes. hg38 coordinates are shown. The blue shading indicates the specific region with an H3K18la increase in PCOS. Exact $P$ values: **b** $P = 5.64 \times 10^{-5}$; **e** $P = 3.29 \times 10^{-4}$. $P$ values were determined by two-tailed unpaired Student's $t$ test, and data are presented as means ± s.e.m. For **a, k**, and **i**, images are representative of four independent biological replicates. Scale bar: 100 μm. Source data are provided as a Source data file.

dysfunction may also contribute to PCOS infertility[3,6,42,43]. For instance, the endometrium of women with PCOS has been reported to have a higher proportion of epithelial cells, indicating the epithelium as a main target of PCOS-related abnormalities, which might adversely affect implantation[42]. However, conclusive clinical evidence is still lacking to confirm the deleterious impact of endometrial dysfunction on the decline in fertility of PCOS[3]. To address this question, we conducted a retrospective cohort study and determined a strong association between endometrial dysfunction and the lower implantation rate in PCOS by rigorously controlling embryonic factors and other potential confounders.

As a large-scale retrospective cohort study involving 4278 participants, we ensured statistical power and sample representativeness, unlike many published studies that suffer from small sample sizes or limited subpopulations[8,44,45]. The clinical pregnancy and ongoing pregnancy rates were lower in high-responder patients with PCOS compared with high-responder patients without PCOS after transferring embryos with comparable euploidy rates, which was not consistent in women with a moderate oocyte yield[45]. Another cohort study, which transferred a single, high-quality euploid blastocyst selected via pre-implantation genetic diagnosis and adjusted for confounders, found significantly higher early miscarriage rates and lower live birth rates in the PCOS group compared to controls. This study focused exclusively on individuals with normal BMI[8]. A large-scale cohort study analyzed 7678 cycles with multivariate adjustments for embryo quality and confounders and reported higher rates of implantation, clinical pregnancy, and live birth in the PCOS group compared to controls[9]. This controversial conclusion might be due to its exclusion criteria, which did not exclude conditions like endometriosis or uterine abnormalities that could affect outcomes. Therefore, our cohort study represents an attractive dataset that can provide direct and rigorous support for endometrial dysfunction as an important contributor to PCOS infertility.

To investigate mechanisms underlying endometrial dysfunction in PCOS, we conducted a comprehensive analysis across multiple models, including clinical samples, Ishikawa cells, and PCOS mice, to draw robust conclusions, considering that the endometrial/uterine physiology is not identical between humans and mice. The human decidua forms routinely and sheds in the absence of an embryo, whereas decidualization of stromal cells occurs after successful embryo implantation in mice[46]. We found that women with PCOS have impaired endometrial receptivity and excessive endometrial ERα, which were also confirmed in PCOS mice. Estrogen and progesterone exert both synergistic and opposing effects on endometrial receptivity, with estrogen playing a dominant role. Excessive estrogen can prematurely close WOI and impair fertility[14]. In our cohort study, women with PCOS had significantly higher estradiol levels at baseline and on the hCG injection day. Together with elevated endometrial ERα, the increased estradiol level is also likely to be responsible for endometrial dysfunction in PCOS.

Our next question is how ERα overexpression affects endometrial/uterine receptivity in PCOS. The estrogen signaling pathway plays a pivotal role in a series of important biological processes, including the regulation of epigenetic modifications[47,48], which are important for endometrial receptivity and are frequently dysregulated in PCOS[49]. Lactylation, a lactate-induced histone modification, plays a significant role in reproductive health, including gametogenesis, embryogenesis, endometrial decidualization, and placentation[27,50–52]. It has been reported that murine uterine decidual cells exhibit increased Kla during mid-pregnancy, while inhibition of lactylation results in impaired uterine decidualization and pregnancy failure[20]. However, excessive endometrial Kla is also harmful. Along with excessive ERα, we also observed elevated H3K18la in the endometrium of PCOS patients and the uterus of PCOS mice. ERα overexpression increases H3K18la via upregulating LDHA/LDHB in Ishikawa cells, and H3K18la, in turn, promotes ERα expression upon its increase. Notably, inhibiting ERα or lactate production restores uterine receptivity in PCOS mice, with improved implantation rates. Thus, the Kla level must be appropriately maintained to ensure endometrial receptivity and implantation. Our observations yield novel insights into the pathogenesis of endometrial dysfunction in PCOS and provide additional therapeutic targets for PCOS infertility.

Recent studies highlight the impact of metformin in restoring endometrial health in patients with PCOS[42,53]. Along with our findings, metformin may rescue endometrial function in PCOS via regulating ERα. It has been reported that metformin can reduce ERα expression in the endometrium of diabetic women with endometrial cancer and Ishikawa cells[54,55]. However, as a classic antidiabetic drug, metformin inhibits mitochondrial oxidative phosphorylation, potentially leading to lactate accumulation[56–59]. In addition, metformin ameliorates liver fibrosis in mice by enriching *Lactobacillus* sp. MF-1 in the gut microbiota[60], and emerging evidence suggests that the gut microbiota and its derived metabolites may influence the microenvironment of the female reproductive tract[61], which could potentially impact endometrial lactate levels. Thus, the interplay between ERα and Kla is complex during the pathogenesis and treatment of PCOS. Extra research efforts are needed to understand how metformin improves endometrial receptivity via regulating ERα/Kla in PCOS.

When comparing endometrial samples from women with PCOS and controls, BMI must be taken into account. Even though the BMIs of PCOS donors for our in vitro studies were within the normal BMI category, the Asian ethnicity experiences insulin resistance with lower BMI, which might obscure the conclusion[62]. To confirm our findings as PCOS-dependent rather than BMI-dependent, PSM was applied to donors in the PCOS and control groups to make the two groups' BMIs comparable. The excessive ERα and H3K18la were still observed in PCOS after PSM (Supplementary Data 1). We also collected additional samples, confirming that ERα and H3K18la are consistently higher in PCOS than in controls across different BMI categories (Supplementary Data 1).

In summary, this study confirms that abnormal endometrial receptivity is a significant contributor to PCOS infertility. We have elucidated a molecular mechanism in which ERα and Kla interactively regulate each other, and their upregulation is associated with impaired endometrial receptivity in PCOS. This study not only enhances our understanding of PCOS infertility but also provides a theoretical basis for optimizing treatment protocols in clinical practice, which is promising for improving the fertility of women with PCOS.

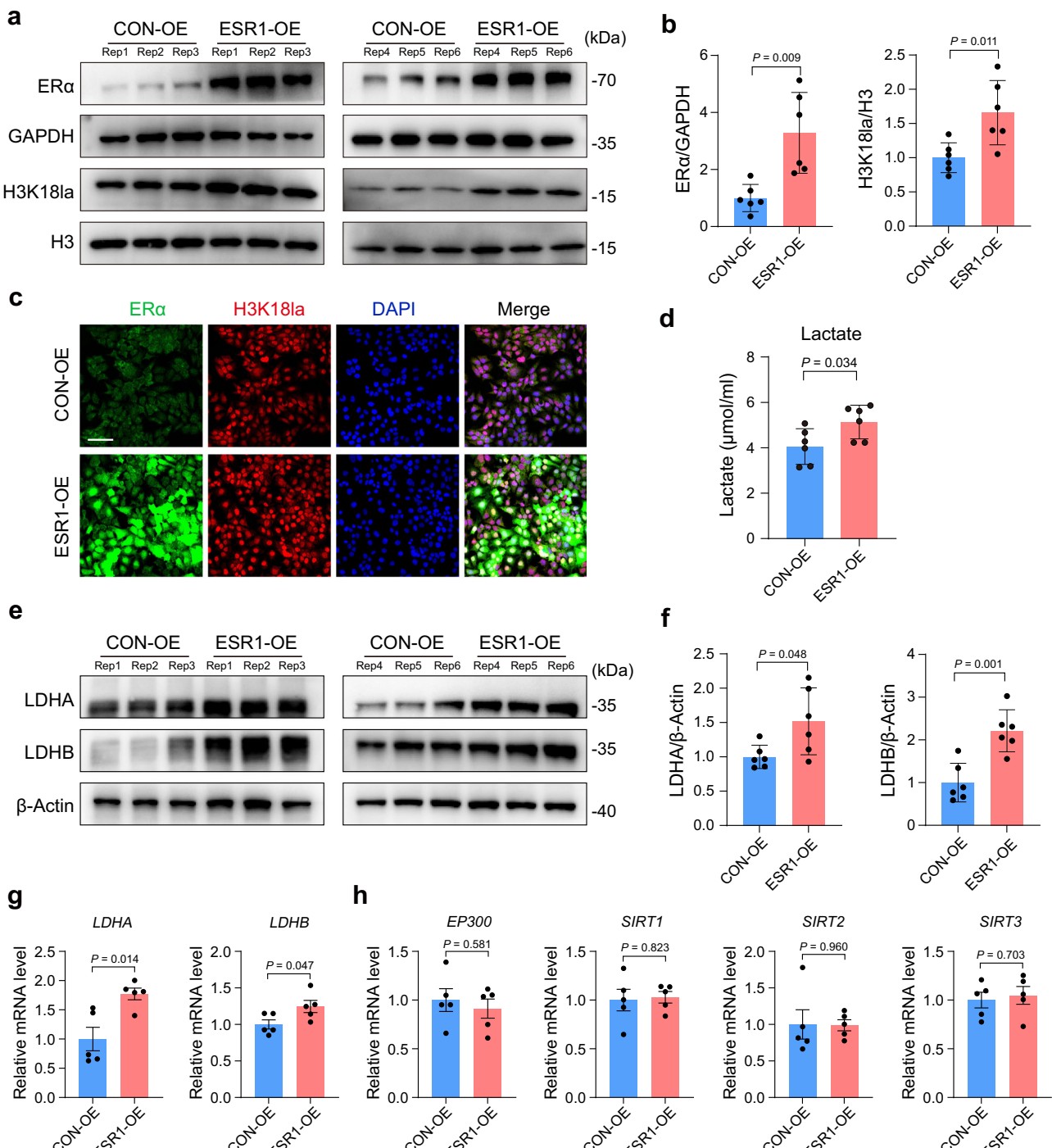

**Fig. 3 | ERα overexpression upregulates Kla in Ishikawa cells. a**, **b** Protein levels of ERα and H3K18la in Ishikawa cells (*n* = 6, biologically independent samples). **c** ERα and H3K18la IF staining in Ishikawa cells. Images are representative of four independent biological replicates. Scale bar: 100 μm. **d** Intracellular lactate levels in Ishikawa cells (*n* = 6, biologically independent samples). **e**, **f** Protein levels of LDHA and LDHB in Ishikawa cells (*n* = 6, biologically independent samples). **g** Relative mRNA levels of *LDHA* and *LDHB* in Ishikawa cells (*n* = 5, biologically independent samples). **h** Relative mRNA levels of *EP300*, *SIRT1*, *SIRT2*, and *SIRT3* in Ishikawa cells (*n* = 5, biologically independent samples). *P* values were determined by two-tailed unpaired Student's *t* test, and data are presented as means ± s.e.m. CON-OE, control; ESR1-OE, *ESR1* overexpression. Source data are provided as a Source data file.

## Methods

### Clinical study design and participants

This study was conducted in compliance with all relevant ethical regulations and was approved by the Ethics Committee of Peking University Third Hospital (Approval No. 2023-190-01). The retrospective cohort study enrolled 4278 patients (aged ≤ 40 years) who underwent IVF/ICSI treatments at the Reproductive Medicine Center of Peking University Third Hospital. The patients were divided into a PCOS group (*n* = 1649) and a control group (*n* = 2629) based on their medical records from January 2018 to December 2021. To address potential confounding, we performed PSM using covariates selected based on established associations with reproductive outcomes in PCOS and infertility research. These included age of females, BMI, infertility duration, miscarriage history, and age of males, all of which are associated with pregnancy

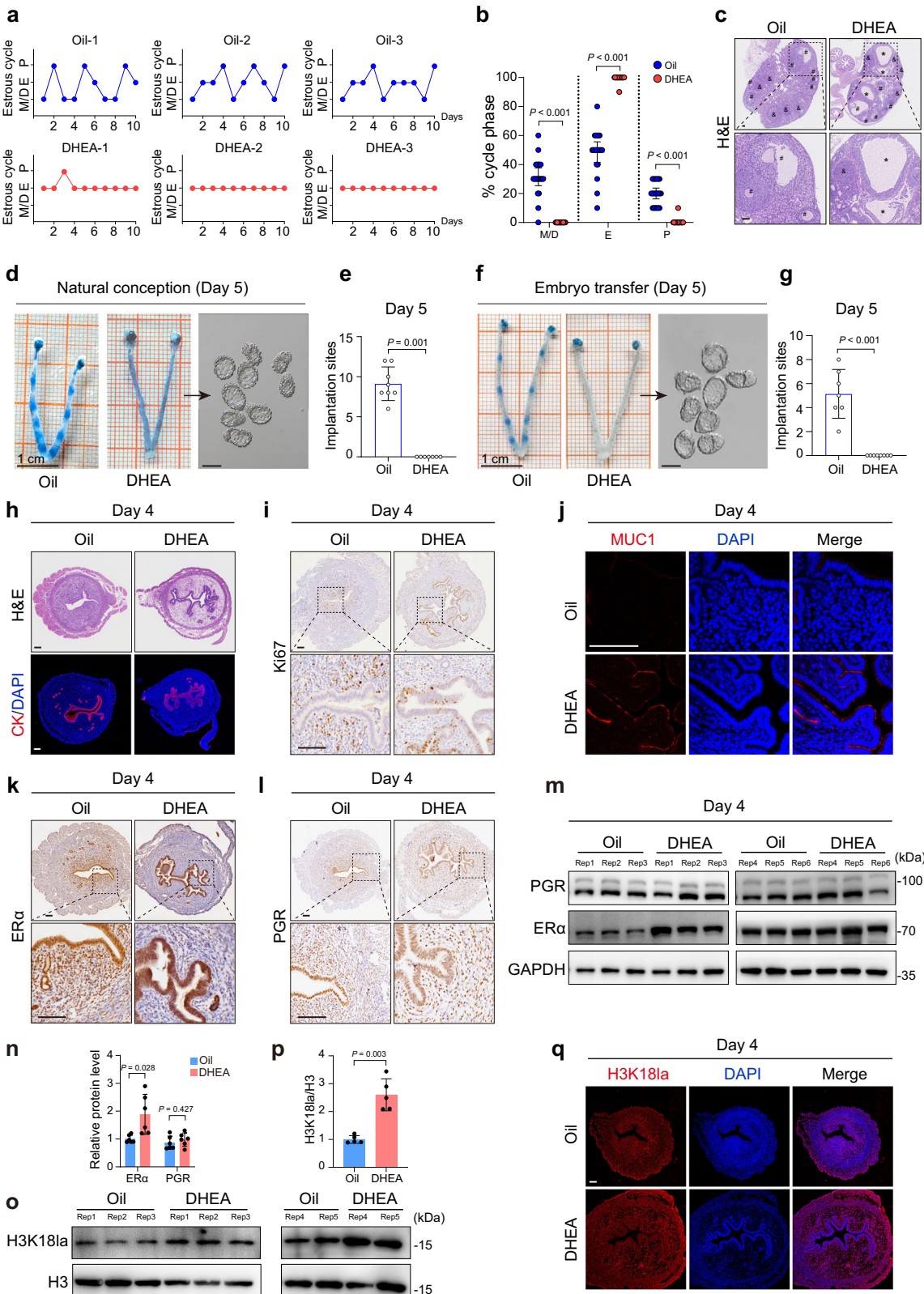

success or live birth rates[63–67]. We excluded variables lying on the causal pathway between PCOS (exposure) and reproductive outcomes (e.g., AMH, AFC) to prevent over-adjustment. Ultimately, 1406 women with PCOS and 1406 controls were included in this analysis. PCOS was diagnosed according to the 2003 Rotterdam criteria.

Inclusion criteria: Women who underwent a frozen-thawed cycle at our hospital from 2018 to 2022, with only one high-quality blastocyst transferred. Exclusion criteria: (1) Presence of endometriosis, recurrent implantation failure (RIF) or recurrent spontaneous abortion (RSA), (2) Thyroid disease, (3) Severe intrauterine adhesion, (4) Severe uterine abnormality. Endometriosis is a chronic inflammatory condition characterized by the presence of endometrial-like tissue (glands and stroma) outside the uterine cavity, primarily affecting pelvic organs (e.g., ovaries and peritoneum), often leading to pain and/

**Fig. 4 | PCOS mice exhibit excessive uterine ERα and H3K18la along with abnormal endometrial receptivity. a** Representative estrous cycles of Oil and DHEA mice. **b** Quantitative analysis of estrous cycles in Oil and DHEA mice ($n = 20$, biologically independent mice). M/D Metestrus/Diestrus, E Estrus, P Proestrus. **c** Hematoxylin and eosin (H&E) staining of ovaries in Oil and DHEA mice. *: cystic follicle; #: antral follicle; &: corpora lutea. **d** ISs visualized by blue dye in naturally conceived Oil mice, and unimplanted embryos in DHEA mice. Black arrows indicate embryos. **e** The number of ISs on day 5 in naturally conceived mice ($n = 8$ for Oil and $n = 7$ for DHEA). **f** ISs visualized by blue dye in embryo-transferred Oil mice, and unimplanted embryos in DHEA mice. Black arrows indicate embryos. **g** The number of ISs on day 5 in embryo-transferred mice ($n = 7$ for Oil and $n = 8$ for DHEA). **h** H&E and rhodamine-phalloidin staining of the cytoskeleton in the murine uterus on day

4. **i** Ki67 IHC staining in the murine uterus on day 4. **j** MUC1 IF staining in the murine uterus on day 4. **k, l** ERα, and PGR IHC staining in the murine uterus on day 4. **m, n** Protein levels of ERα and PGR in the murine uterus on day 4 ($n = 6$, biologically independent mice). **o, p** H3K18la levels in the murine uterus on day 4 ($n = 5$, biologically independent mice). **q** H3K18la IF staining in the murine uterus on day 4. Exact $P$ values: **b**: M/D, $P = 2.02 \times 10^{-9}$; E, $P = 1.45 \times 10^{-11}$; P, $P = 3.26 \times 10^{-4}$; **g**: $P = 3.11 \times 10^{-4}$. For **n, p**, $P$ values were determined by two-tailed unpaired Student's $t$ test, and data are presented as means ± s.e.m. For **b, e, g**, $P$ values were determined by two-tailed unpaired Mann–Whitney $U$ rank-sum test, and data are presented as medians with interquartile ranges. For **c, h–l, q**, images are representative of four independent biological replicates. Scale bar: 100 μm. Source data are provided as a Source data file.

or infertility[68]. RIF is defined as the failure to achieve a clinical pregnancy after at least three transfer cycles involving ≥4 high-quality embryos[69]. RSA refers to two or more consecutive pregnancy losses before 28 weeks of gestation with the same partner, including biochemical pregnancies.

Embryo selection: High-quality blastocysts were evaluated using the Gardner criteria observed on days 5–6; Blastocysts graded ≥3BB were considered high-quality. The primary outcome, implantation rate, was defined as the presence of at least one gestational sac detected on vaginal ultrasound 4–5 weeks after embryo transfer, regardless of intra- or extrauterine location. The presence of one or two gestational sacs was considered a single successful implantation. Secondary pregnancy outcomes included positive pregnancy, chemical pregnancy, miscarriage, premature delivery, gestational age at delivery, birthweight, and live birth. Positive pregnancy was defined as a serum β-hCG level ≥ 5 U/L on the 14th day after embryo transfer, while chemical pregnancy was defined as a serum β-hCG level ≥ 10 U/L on the same day. Miscarriage was categorized as the spontaneous loss of pregnancy before 28 weeks of gestation. Premature delivery was defined as the birth of a viable newborn between 28 and 37 weeks of gestation. Live birth was defined as the delivery of a viable newborn at or beyond 24 weeks of gestation.

## Patient samples and ethical approval
All endometrial samples were obtained with the patients' informed consent, and the study was approved by the Ethics Committee of Peking University Third Hospital (Approval No.2023-192-02 and No.2023-718-01). According to the inclusion and exclusion criteria, volunteers were recruited and provided informed consent. Endometrial tissues were collected at different stages of the menstrual cycle, determined by the last menstrual period, serum LH peak levels, and confirmation of ovulation by transvaginal ultrasound. The proliferative phase, menstrual cycle days 5–14; the early secretory phase, 1–4 days post-ovulation; the mid-secretory phase, 5–8 days post-ovulation; the late secretory phase, 9–14 days post-ovulation. The early pregnancy status was confirmed by ultrasonographic visualization of an intrauterine gestational sac. The phase of endometrial samples was further confirmed by two senior pathologists using examination of Hematoxylin and Eosin (H&E) stained sections. Patient details are provided in Supplementary Table 2. Endometrial samples collected in this study are stored at −80 °C and are available upon request from the corresponding author. This study does not involve any human embryos, gametes, or stem cells.

## Isolation, culture, and treatment of human endometrial cells
The isolation and culture of primary endometrial epithelial cells and stromal cells were performed according to previous studies[70]. After one passage, decidualization was induced in the isolated stromal cells. The medium was changed to phenol red-free DMEM/F12 supplemented with 2% (v/v) CS-FBS, and 1 μM MPA (HY-B0469, MCE) and 0.5 mM cAMP (S7857, Selleck) were added for 6 days. When

the cell density reached 50% to 60%, the medium was replaced every 48 h.

Ishikawa cells (CL-0283) were purchased from Wuhan Punosai Life Science and Technology Co., Ltd. (Wuhan, China) and cultured in medium containing 10% (v/v) fetal bovine serum and 0.5%(v/v) penicillin-streptomycin, incubated at 37 °C with 5% $CO_2$. Ishikawa cells were regularly authenticated by morphological observation and short tandem repeat (STR) analysis as well as tested for the absence of mycoplasma contamination (40618ES25, Yeasen). The cells were treated with two reagents: NaLa (71718, SIGMA) and Oxamate (O2751, SIGMA).

## Human endometrial organoid generation
Endometrial tissues were minced into small fragments under sterile conditions and subjected to enzymatic digestion in 5 ml PBS containing 1 mg/ml collagenase IV (Sigma-Aldrich, USA, C4-28) at 37 °C for 20–30 min with gentle agitation. Digestion was terminated by adding an equal volume of advanced DMEM/F-12 (Gibco, USA, 12634028). The suspension was gently pipetted and sequentially filtered through 100 μm (Falcon, USA, 352360) and 40 μm (Falcon, USA, 352340) cell strainers. Glandular fragments were collected via backwashing of the 40 μm strainer, resuspended in ice-cold Matrigel (Corning, USA, 536231), and plated into 48-well plates. After polymerization at 37 °C, the Matrigel domes were overlaid with 250 μl of organoid expansion medium. The medium consisted of advanced DMEM/F-12 supplemented with 1% N2 (ThermoFisher, USA, 17502048), 1% B27 (ThermoFisher, USA, 17504044), 1% Glutamax (Gibco, USA, 35050061), 500 ng/ml R-Spondin 1 (Novoprotein, China, CX83), 100 ng/ml Noggin (Novoprotein, China, CB89), 1.25 mM N-Acetylcysteine (Sigma-Aldrich, USA, A0737), 10 mM Nicotinamide (Sigma-Aldrich, USA, N0636), 25 ng/ml HGF (Novoprotein, China, CJ72), 50 ng/ml EGF (Novoprotein, China, C029), 100 ng/ml FGF10 (Novoprotein, China, CR11), 10 μM Y-27632 (MCE, USA, HY-10583), 500 nM A83-01 (MCE, USA, HY-10432), 1× ITS (ThermoFisher, USA, 41400045), and 1% Penicillin/Streptomycin (Gibco, USA, 15140122). The medium was refreshed every 2–3 days. Organoids were passaged every 7–10 days by mechanical disruption and re-embedded in Matrigel. For subsequent analyses, organoids were harvested using cell recovery solution (Corning, USA, 354253).

## Animal feeding and treatment
The study was approved by our Institutional Animal Ethics Committee of Peking University Third Hospital (Approval No. A2022111). C57BL/6J female mice at 3 weeks of age were obtained from Beijing Vital River Laboratory Animal Technology (Beijing, China). All mice had access to food and water and were raised in a controlled environment with a 12-h light/dark cycle, at a room temperature of 20–25 °C, and with a humidity of 55% ± 10%. According to previous literature reports[41], 3-week-old C57BL/6J female mice were given a daily subcutaneous injection of DHEA (6 mg/100 g body weight; D4000-10 g, Sigma Aldrich, USA) dissolved in 0.2 mL of sesame oil for 21 days to establish the PCOS mouse model, whose body weights were recorded every

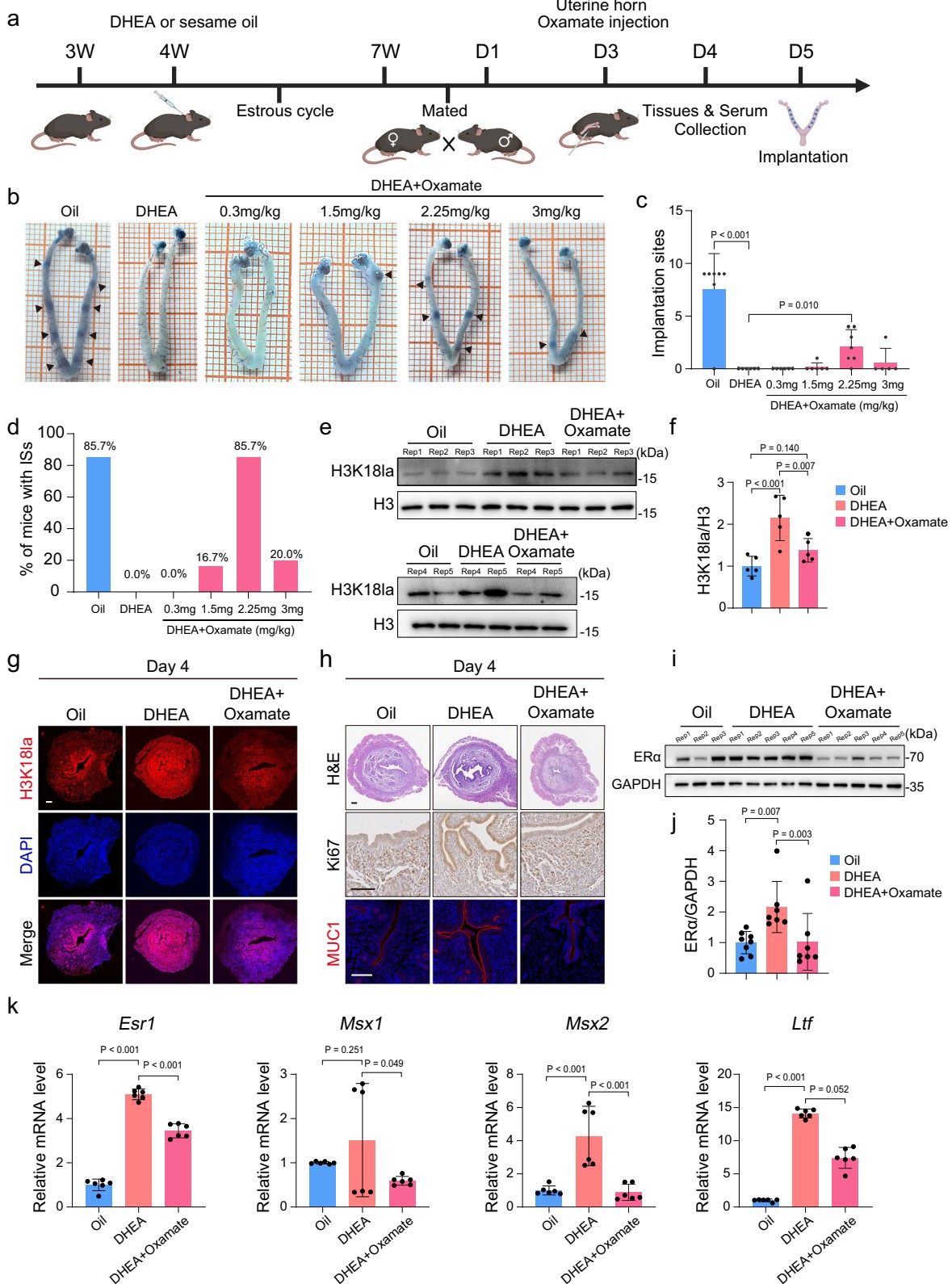

other day during treatment. From day 11 of injection, estrous cycles were monitored daily by vaginal cytology. At the study endpoint, glucose tolerance test (GTT) and insulin tolerance test (ITT) were performed, followed by orbital blood collection for serum hormone measurements (testosterone [T], estradiol [E2], and luteinizing hormone [LH]). Ovaries were processed for histopathological examination with hematoxylin and eosin (H&E) staining to evaluate cystic follicle

formation and corpora lutea counts for validation of a PCOS-like phenotype.

Embryo transfer experiment. Preparation of donor mice: Female C57BL/6J mice at 8 weeks of age were accommodated for 5–7 days before receiving an intraperitoneal (i.p.) injection of 5 IU PMSG (AC13853-5*1000 IU, Acmec). Following this, they received i.p. injections with 5 IU hCG (C79010-1mg, Acmec) after 46–48 h and mated

**Fig. 5 | Kla reduction recovers abnormal uterine receptivity in PCOS mice.**
**a** Schematic diagram of PCOS mice construction and drug intervention. Created in BioRender. Hongying, S. (2025) https://BioRender.com/nwdxi9x. **b** ISs visualized by blue dye in naturally conceived mice. Black arrows indicate ISs. **c** The number of ISs in naturally conceived mice on day 5 (Oil, $n = 7$; DHEA, $n = 6$; DHEA + 0.3 mg/kg Oxamate, $n = 6$; DHEA + 1.5 mg/kg Oxamate, $n = 6$; DHEA + 2.25 mg/kg Oxamate, $n = 7$; DHEA + 3 mg/kg Oxamate, $n = 5$; biologically independent mice). **d** Percentage of naturally conceived mice with ISs on day 5. **e, f** H3K18la levels in the murine uterus on day 4 ($n = 5$, biologically independent mice). **g** H3K18la IF staining in the murine uterus on day 4. **h** H&E, Ki67 IHC, and MUC1 IF staining in the murine uterus on day 4. **i, j** Protein levels of ERα in the murine uterus on day 4 ($n = 8$ for Oil, $n = 7$ for DHEA and DHEA + Oxamate, biologically independent mice). **k** Relative mRNA

levels of *Esr1*, *Msx1*, *Msx2*, and *Ltf* in the murine uterus on day 4 ($n = 6$, biologically independent mice). Exact $P$ values: **c** Oil vs. DHEA, $P = 2.33 \times 10^{-4}$; **f** Oil vs. DHEA, $P = 4.23 \times 10^{-4}$; **k** For relative mRNA levels of *Esr1*, Oil vs. DHEA, $P = 7.84 \times 10^{-14}$, DHEA vs. DHEA + Oxamate, $P = 2.99 \times 10^{-8}$; for relative mRNA levels of *Msx2*, Oil vs. DHEA, $P = 9.85 \times 10^{-5}$, DHEA vs. DHEA + Oxamate, $P = 6.99 \times 10^{-5}$; for relative mRNA levels of *Ltf*, Oil vs. DHEA, $P = 9.89 \times 10^{-5}$. For **f** and relative mRNA levels of *Esr1*, *Msx1*, *and Msx2* in (**k**), $P$ values were determined by one-way ANOVA followed by Tukey's post hoc test, and data are presented as means ± s.e.m. For **c, j** and relative mRNA levels of *Ltf* in (**k**), $P$ values were determined using Kruskal–Wallis test followed by Dunn's post hoc test, and data are presented as medians with interquartile ranges. For **g, h**, images are representative of four independent biological replicates. Scale bar: 100 μm. Source data are provided as a Source data file.

with fertile male mice in a 1:1 ratio. Due to the difficulty in mating with PCOS model mice, they were also induced to estrus using the same medication protocol. Meanwhile, recipient mice were mated with vasectomized male mice. Vaginal plugs were checked the next morning between 8:00 and 10:00, and the day of plug detection was designated day 1. On day 4 between 8:00 and 10:00, five high-quality blastocysts were transferred into each horn under anesthesia. Implantation sites (ISs) were identified by intravenous injection of Chicago blue dye solution (C8679, Sigma-Aldrich), and the number of implantation sites, marked by distinct blue stripes, was recorded on day 5. To confirm pregnancy in plug-positive females without visible implantation sites, the uterine horns were rinsed in M2 medium (M1250, Yuabio) and examined for the presence of blastocysts.

On day 3, PCOS mice were randomly allocated into groups. Each uterine horn was injected with Oxamate (0.3–3 mg/kg) or MPP (8.14 μg/kg). Mice were euthanized by cervical dislocation after sedation for sample collection. Uterine tissue samples were collected on day 4 for subsequent molecular analysis and on day 5 for counting ISs.

## Western blotting
Samples were homogenized in RIPA lysis buffer containing protease inhibitor, and proteins were separated on a 4–20% gradient SDS-PAGE gel. Following this, the proteins were transferred to PVDF membranes (Millipore) and then blocked with 5% (w/v) non-fat dry milk for about 1 h. Specific primary and secondary antibodies were then sequentially incubated with the membranes. The blots were visualized with an enhanced chemiluminescent substrate (Thermo Fisher Scientific) and quantified with ImageJ, normalized to GAPDH, β-Actin, H3, or H4. Details of antibodies are summarized in Supplementary Table 3.

## Quantitative real-time PCR
Total RNA was extracted from uterine tissues or cells using TRIzol® reagent (15596018, Thermo Fisher). RNA purity was assessed using the ND-1000 Nanodrop. Each RNA sample had an A260:A280 ratio above 1.8 and an A260:A230 ratio above 2.0. cDNA was synthesized using the PrimeScript RT reagent kit (FSQ-101, Toyobo) according to the manufacturer's instructions. Quantitative real-time PCR was performed using PowerUp SYBR Green Master Mix (A25742, Thermo Fisher) on a QuantStudio 12K Flex system (Applied Biosystems). Every PCR experiment was repeated at least five times, with GAPDH used as the housekeeping gene for human samples and Tbp for mouse samples. The relative gene expression levels were calculated using the $2^{-\Delta\Delta Ct}$ method. PCR primers are listed in Supplementary Table 4.

## Detection of L-lactate content
The lactate content in endometrial tissues and cells was measured using the Lactate Assay Kit (BC2230, Solarbio) following the manufacturer's instructions. Briefly, tissues and cells were collected, lysed, and sonicated on ice, then centrifuged at $12,000 \times g$ at 4 °C for 20 min. The supernatants were collected for lactate measurement. Next, the spectrophotometer was preheated to a stable state, and the standards

were prepared at various concentrations. Samples, standards, distilled water, and various reagents were sequentially added, thoroughly mixed, and incubated at 37 °C for 20 min. After the reaction, the samples were centrifuged to remove impurities from the supernatant. Finally, the absorbance at 570 nm was measured for each sample, and lactate levels were calculated from the standard curve.

## Immunostaining
Tissue sections of 5 μm thickness were cut from formalin-fixed, paraffin-embedded human endometrial and mouse uterine tissues. The samples were deparaffinized and treated for antigen retrieval by boiling in citrate buffer for 15–20 min to repair antigens, followed by blocking with 0.5% BSA-PBS for 1 h. For immunohistochemistry (IHC), the following antibodies were used: ERα, PGR, Ki67, COX2, and HOXA10. After overnight incubation with primary antibodies, signals were displayed using horseradish peroxidase-conjugated secondary antibodies. Additionally, immunofluorescence (IF) was also performed on paraffin-embedded sections and cells. These were incubated with Pan Kla, H3K18la, H4K12la, ERα, MUC1, CK, Vimentin, and F-actin (phalloidin) primary antibodies. Signals were visualized with Alexa Fluor 488 (anti-rabbit, Invitrogen) and Alexa Fluor 555 (anti-mouse) secondary antibodies, and nuclei were stained with DAPI. Detailed antibody information is available in Supplementary Table 3. For IF, images were captured using a Zeiss LSM 880 confocal microscope with Airyscan super-resolution capability. For IHC, images were acquired with a scanner WS-10 (WISLEAP, China) and analyzed with ImageJ.

## Plasmids and cell transfection
Ishikawa cells were seeded in 6-well plates at a concentration of $2 \times 10^5$ cells per well and cultured to a confluence of approximately 60%, then transfected with the lentiviral control virus HBLV-ZsGreen-PURO or with the target virus HBLV-ESR1-3XFLAG-ZsGreen-PURO (Hanbio Biotechnology) using Lipofectamine 3000 (Invitrogen) according to the manufacturer's instructions. The sequences of cloning primers used in this study are listed as follows: Forward, 5'- tactagaggatctatttccggtGaattcGCCACCATGACCATGACCCT-3'; Reverse, 5'-AGTCACTTAAGCTTGGTACCGAggatccGACCGTGGCAGGGAAACCCT-3'. After 48 h of infection, cells that were not infected efficiently were killed by adding and maintaining 1.0 μg/mL puromycin. In this way, a stable *ESR1*-overexpressing cell line, maintained under puromycin selection, was finally obtained.

## CUT&Tag library preparation
According to the manufacturer's instructions, endometrial cell nuclei from the PCOS and control groups were isolated for the CUT&Tag assay using the Cell Nucleus Isolation Kit (Cat. No. 52201-10, Bioyou). Assays were performed using the Illumina Hyperactive CUT & Tag Kit (cat. no. TD-903, Vazyme Biotech). Concanavalin A (ConA)-coated magnetic beads were added and incubated at room temperature. The nonionic detergent digitonin was used to permeabilize membranes. The nuclei were subsequently incubated with primary antibodies targeting H3K18la, followed by secondary antibodies and pA-Tn5.

Utilizing this transposase, the DNA fragments bound to the specific proteins were precisely excised. Following cleavage, the DNA fragments were ligated to N5 and N7 adapters and subsequently amplified using N5 and N7 primers. Sequencing of the libraries was performed on the Illumina NovaSeq 6000 platform to generate 150 bp paired-end reads for analysis.

## CUT&Tag data analysis

Clean reads were generated from the raw sequence data using fastp. Clean reads were then aligned to the human reference genome (hg38) using Bowtie2. Peak calling was conducted with MACS2, using narrowPeak settings. Sequences within a 400 bp window were extracted, and those containing significant repetitive elements, as identified by RepeatMask, were excluded. Peaks were visualized using IGV. ChIP-seeker was employed to annotate peaks and retrieve gene-related information. Motif analysis was performed using MEME and DREME, and identified motifs were compared to the known motif database. Deeptools was used to visualize and compare peaks across different groups. For H3K18la CUT&Tag data, differential analysis was performed between control ($n = 4$) and PCOS ($n = 4$) groups with Diffbind. Differential binding peaks were identified as having $P$-adjusted values < 0.05 and fold changes >2 or <0.5.

## Statistical analysis

This retrospective cohort study encompassed 4278 subjects. To adjust for confounders, PSM was employed for variables including the female age, history of miscarriage, infertility duration, BMI, and male age, maintaining a 1:1 matching ratio between the PCOS and control groups, with a caliper of 0.02. We used data after PSM for the primary analysis to evaluate the robustness of our findings, while the data before PSM were used for sensitivity analysis. For parametric tests, two-tailed unpaired Student's $t$ test was used to evaluate the statistical significance between two groups, and one-way ANOVA followed by Tukey's post hoc test was used for three or more groups. Data are shown as the means ± s.e.m./SD. For non-parametric tests, data are presented as medians [Q1, Q3]. $P$ values were determined by two-tailed unpaired Mann–Whitney $U$ rank-sum test between two groups, Kruskal–Wallis test followed by Dunn's post hoc test for three or more groups, and chi-square test or Fisher's exact test for categorical variables. Statistical analyses were performed using R (version 4.2.2) and SPSS (version 26.0) software. A $P$ value < 0.05 was considered statistically significant.

## Reporting summary

Further information on research design is available in the Nature Portfolio Reporting Summary linked to this article.

# Data availability

All data supporting the findings described in this manuscript are available in the article and in the Supplementary Information and from the corresponding author upon request. Source data are provided with this paper. Raw sequencing data of CUT&Tag are available on the Genome Sequence Archive (https://ngdc.cncb.ac.cn/gsa-human/browse/HRA009852). BED files of CUT&Tag are available on the Open Archive for Miscellaneous Data (https://ngdc.cncb.ac.cn/omix/release/OMIX012938). The human reference genome hg38 is available on the National Center for Biotechnology Information (https://ftp.ncbi.nlm.nih.gov/genomes/all/GCF/000/001/405/GCF_000001405.39_GRCh38.p13/). Source data are provided with this paper.

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

## Acknowledgements

This work was supported by the National Key R&D Program of China (2022YFC2702500, 2021YFC2700601), National Natural Science Foundation of China (82288102, 82271699, 82560305, 82301888, 82225019, 82192873), the Key Clinical Projects of Peking University Third Hospital (BYSYZD2023028), the Beijing Nova Program (20220484073), the "Tianshan Talent" Program of Xinjiang (TSYC202401B074), and Science and Technology Program of XPCC (2023ZD008). We would like to thank all the patients and staff of the Centre for Reproductive Medicine, Department of Obstetrics and Gynecology, Third Hospital of Peking University, for their contributions. We thank CapitalBio Corporation and Annoroad for technical assistance.

## Author contributions

Concept and design: H.P., R.L., Y.Y., P.Z., and H.S. Acquisition, analysis, or interpretation of data: H.S., Y.W., B.L., K.H., and C.X. Drafting of the manuscript: H.S., Y.W., and P.Z. Collection of the clinical data: X.C., Z.Y., F.L., T.P., F.D., and M.L. Critical revision of the manuscript for important intellectual content: H.P., H.S., and Y.W. Administrative, technical, or material support: H.P., R.L., and Y.Y. Supervision: H.P., and R.L. All authors have approved the final version for submission.

## Competing interests

The authors declare no competing interests.
