## [Transparent Peer Review File · Nature Communications]

Women with polycystic ovary syndrome exhibit impaired endometrial receptivity with excessive ER α and histone lactylation

Corresponding Author: Professor Heng Pan

Version 0:

Reviewer comments:

Reviewer #1

(Remarks to the Author)

Comments to the Authors:

In the manuscript, the authors investigated the link between excessive ER α /histone lactylation and impaired endometrial receptivity in women with polycystic ovary syndrome (PCOS). These data provide new perspectives on the potential therapeutic strategy for endometrial dysfunction. This article represents an interesting contribution to the field, as the work presented is original, and the content is significant. Nevertheless, the article requires major revisions, as the current version is not suitable for publication. Several queries need to be addressed, and corrections are necessary. Providing additional details in the methods section, as well as supplementing and revising the results, would enhance the manuscript's value and comprehensiveness.

Major concerns:

Abstract and Introduction

1. The Authors should explicitly state a research hypothesis. The introduction should conclude with the research hypothesis and the purpose of the study, rather than the obtained results.

Materials and Methods

1. Repeating the experiment two or three times (2n or 3n, where n is the number of each experiment) may not be sufficient for a robust statistical analysis. Therefore, the authors should increase the number of experiments and explain the rationale for the observed variations in n across different experiments.
2. What was the cell viability of the human primary endometrial epithelial and stromal cells?"
3. Detection of L-lactate content:
 - a) Were any controls included to validate the L-lactate detection assay?
 - b) What are the sensitivity and specificity of the assay used to measure L-lactate?
4. Immunostaining: Please provide information on negative controls, species reactivity of antibodies, and the type and brand of the fluorescence microscope used.
5. Plasmids and cell transfection:
 - a) Were any specific conditions or treatments applied to improve transfection efficiency?
 - b) What was the cell viability post-transfection?
6. Western blot: Please provide details on negative controls, species reactivity of the antibodies, and the method used for analysing the results.
7. Quantitative Real-Time PCR (qRT-PCR):
 - a) Please adhere to MIQE guidelines.
 - b) A validation experiment is necessary to confirm the accuracy of the delta-delta CT calculation. What were the absolute values of the slope of delta CT vs. log input?
 - c) Using a single reference gene for qRT-PCR analysis is insufficient. Additionally, the stability of the reference gene in this

study should be assessed.

d) What negative controls were included in qRT-PCR? Were the primers designed to overlap exon-exon boundaries? How was potential DNA contamination controlled?

e) Providing a detailed description of qRT-PCR conditions in a table format would improve clarity. This should include primer sequences, primer concentrations, PCR amplicon lengths and GenBank accession numbers.

8. CUT&Tag assay:

a) What was the starting number of cells before nuclei isolation, and how many nuclei were obtained per sample?

b) How did you assess the effectiveness of the antibodies used?

c) What negative and positive controls were included in the analysis?

Results, Figures and Conclusions

1. To properly assess decidualisation, protein levels of decidual markers, such as PRL and IGFBP1, should be measured, rather than relying solely on their gene expression levels.

2. Immunostaining: Figures should include images of negative controls.

3. Immunostaining analysis primarily serves a qualitative purpose, confirming protein presence and localisation. The conclusions drawn from these results should not be based solely on the expression levels inferred from immunostaining.

4. Western blot figures: Adding a molecular weight marker to the presented blots would facilitate the interpretation of the protein size and improve result analysis.

5. Concerns regarding the reference protein panels in Figure 1C&I (GAPDH), Figure 2E (H3), and Figure 2G (H4): The intensity of these proteins differs between control and PCOS groups. Reference proteins should not show significant variations between samples. Given this discrepancy, statistical analyses should be conducted on these proteins, and the results should be presented.

6. In Figure 2, the text states that experiments included at least three biological replicates, whereas the CUT&Tag methodology description mentions only two replicates. This discrepancy should be clarified.

7. Each figure should be self-explanatory. Abbreviations in figure legends should be defined to ensure that figures can be interpreted independently of the main text.

Reviewer #2

(Remarks to the Author)

Remarks to the Author:

In this manuscript, Shan et al. attempt to define the mechanisms underlying impaired endometrial receptivity in women with PCOS. To answer this question, they use correlative studies in humans and functional tests in mice. The main findings of the study are:

1. Strictly controlling for embryonic factors and potential confounders, a large cohort study shows a lower implantation rate in women with PCOS. This suggests that endometrial abnormalities contribute to the reproductive problems in PCOS and are a major factor in poor pregnancy outcomes.

2. The authors show increased levels of ER α and H3K181a in the mid-secretory endometrium of women with PCOS and also demonstrate that overexpression of ER α increases H3K181a via LDHA/LDHB in Ishikawa cells.

3. Impaired embryo implantation is still observed when blastocysts are transferred from control mice to PCOS mice, suggesting endometrial dysfunction as a possible cause of reproductive failure in PCOS. This defect could be significantly improved by reducing H3K181a in the uterus of PCOS mice, suggesting a potential therapeutic target to improve reproductive outcomes in PCOS.

Overall, the proposed regulatory model is well supported by the mouse data, and the human data provide valuable insights into the mechanisms of endometrial dysfunction in PCOS. The manuscript is well written and well organized. Most of the conclusions are robust and consistent with the results. This work will be of great interest to a wide audience in the field. However, the regulatory relationship between H3K181a and ER α needs further clarification and several important questions need to be addressed before publication in Nature Communications.

1) For the retrospective cohort study:

a) The selection of covariates is a crucial step in PSM as it directly affects the effectiveness of the matching process and the reliability of the subsequent analysis. Please describe the steps and principles that the authors followed in selecting the variables.

b) Elevated BMI is a typical feature of PCOS. Would the inclusion of BMI in the PSM lead to over-matching?

c) Exclusion criteria for the cohort study include endometriosis and RSA. The authors should clarify the diagnostic criteria for endometriosis and RSA.

d) It is mentioned that endometrial tissue will be collected from different menstrual cycles. How are the different menstrual cycles determined?

2) For analysis using the human endometrium:

a. In addition to PGR and HOXA10, were changes in HAND2 expression or activity observed?

b. At what time points were the samples in Fig. 1C collected?

c. Lines 118-120: Please include representative images of untreated cells at day 0 to illustrate the morphological changes between the experimental groups.

d. Line 127: The statement implies that LIF is a downstream gene of PGR. Is there direct evidence to support this? Please clarify this.

e. Fig. 2A-B, I-J: The scale bars in these panels appear to be inconsistent in length, although they are indicated as 100 μ m.

The authors should check the accuracy of the scale bars.

f. Please explain the method used to normalize the CUT&Tag data.

g. Does the inhibition of ER α have an effect on LDHA/LDHB?

h. In addition to cell line experiments, it would be very valuable to include organoids from patient samples. This would provide more physiologically relevant evidence and further elucidate the underlying mechanisms.

3) For the mouse model:

a. LINE179-180: The authors should briefly discuss the differences in endometrial physiology between humans and mice.

b. In Fig. 4, the scale bar is missing in several figures. Please add the scale bar to ensure accurate interpretation of the images.

c. The authors should investigate the functions of stromal cells in PCOS and treated mice, e.g. apoptosis.

d. Endometrial lactate levels are elevated in PCOS patients, which has not been studied in PCOS mice. This is important to confirm the robustness of the finding in different species.

e. While lactate is essential for normal embryo implantation, excessive H3K181a is detrimental in PCOS. The authors should discuss the importance of maintaining adequate intrauterine lactate levels for successful embryo implantation.

f. In PCOS mice, it has not been tested whether inhibition of ER α could restore implantation and reduce H3K181a levels. This is important to confirm the effects of ER α on H3K181a.

Reviewer #3

(Remarks to the Author)

Reviewer #4

(Remarks to the Author)

Shan et al., have assessed the role of androgen (DHEA) exposure on endometrial lactation-driven histone modifications that would also explain the higher ESR α expression and proliferation in PCOS endometrium. Blocking lactation rescued the endometrium from excess estrogen effect.

The idea is interesting and approach is novel, however there are some concerns in the manuscript. The human data is also intriguing with appropriate statistical analysis.

1. The major concern is the poor quality of data and inadequately explained processes and samples.
2. The mouse model is not adequately described. When was the DHEA injection (s) given? It was not explained why hCG was given before mating (lots of background, poor quality samples)
3. The estrous cycles seem odd in DHEA mice, usually the mice are in diestrus doi.org/10.3389/fendo.2022.1030151
4. Some of the results are not clear (ratios etc)
5. The testosterone levels are also surprisingly high in PCOS and control mice.
6. In many experiments only n=3 is stated. More accuracy n is needed.

Version 1:

Reviewer comments:

Reviewer #1

(Remarks to the Author)

Dear Authors,

I would like to thank the authors for their detailed and thoughtful responses to the comments raised during the review process. I have carefully evaluated the revised manuscript and the point-by-point replies. The authors have satisfactorily addressed all major concerns.

Specifically, the revised introduction now clearly articulates the research hypothesis and objectives. The authors have appropriately increased the number of biological replicates and explained the rationale for variation in sample size across experiments. Methodological rigour has been strengthened, including validation of primary cell viability, lactate quantification assays, qRT-PCR compliance with MIQE guidelines, and improved documentation for immunostaining and Western blot analyses.

Concerns regarding reference protein variability, CUT&Tag replicates, and the interpretation of immunohistochemistry have been resolved through repeat experiments and improved clarity in figure legends and supplementary data.

Regarding the authors' query on including multiple normalisation figures for qRT-PCR: In my opinion, presenting one well-validated reference gene per species in the main figures is sufficient. The additional normalisation results included in the

supplementary materials provide adequate transparency for readers.

In summary, I find the revised manuscript to be significantly improved. The authors have demonstrated methodological care, appropriate statistical reasoning, and transparency in reporting. I recommend acceptance of the manuscript.

Reviewer #2

(Remarks to the Author)

The authors have adequately addressed my previous comments, and I find that the manuscript has improved significantly as a result.

Reviewer #3

(Remarks to the Author)

Reviewer #4

(Remarks to the Author)

The paper has improved significantly and the authors have rigorously replied to my concerns and comments. The data provides novel approach to improve endometrial receptivity in women with polycystic ovary syndrome (PCOS) by targeting $E\alpha$ /histone lactylation. I congratulate the team to be able to generate also new data for this manuscript.

Some additional comments :

1. What is the role of BMI? For example the BMIs for the patients donating their endometrium for the in vitro studies differ from the controls. Recent studies imply that BMI also has an effect on the endometrium and epithelial cell driven organoids and may explain some changes between the study groups.
<https://pubmed.ncbi.nlm.nih.gov/40499150/>

Even though the BMIs were inside the normal BMI category the asian ethnicity experience insulin resistance with lower BMI that is most likely the case with the samples also used here.

Could the authors strengthen their results to show PCOS-independent (and not BMI dependent) difference between cases and controls. For example show if the in vitro samples were matched for BMI.

Please give also out the protocol for the organoid generation.

2. Some recent key papers from the field could be cited as they support the endometrial dysfunction in PCOS independently from the ovaries and that epithelium could be main target given the higher proportion of epithelial cells in PCOS endometrium. Also the role of Metformin has also been investigated to "rescue" PCOS endometrium. Can the authors elucidate how metformin treatment would fit in their theory.
<https://pmc.ncbi.nlm.nih.gov/articles/PMC12176659/>
<https://pubmed.ncbi.nlm.nih.gov/40344073/>

3. Supplementary table 1. Please explain PSM for the readers.

Version 2:

Reviewer comments:

Reviewer #4

(Remarks to the Author)

I have no further comments. The authors had sufficiently answered to my questions. They had included some of the recent work, assessed the role of BMI and provided the protocol for EEO generation.

We would like to sincerely thank reviewers for their valuable and constructive comments, which have greatly helped us improve the quality and impact of our manuscript. We have thoroughly revised the manuscript and carefully addressed all raised concerns. Our major revisions include:

1. Increase the sample sizes to enhance the robustness of our conclusions. We have employed at least five biological replicates for most experiments, except for CUT&Tag, which requires a larger sample volume. Nonetheless, four biological replicates in CUT&Tag are still larger or at least comparable to published studies.
2. Enhance the data quality by rigorously following the MIQE guidelines in qRT-PCR experiments, including extra reference genes for qRT-PCR, maintaining stable levels of reference proteins in western blots (WB), presenting summarized statistical results when necessary, and implementing proper positive/negative controls for experiments.
3. Improve the clarity of our manuscript by adding detailed information for each experiment, such as a detailed description of sample collection, model establishment, inclusion/exclusion criteria of the cohort study, sample sizes, and agent specifications.
4. Include further experiments in cell lines, patient-derived organoids, and the mouse model to support our findings and further confirm the regulatory relationship among ER α , histone lactylation, and endometrial receptivity.

We have also prepared a detailed point-by-point response to address each reviewer's comment. We hope the revised manuscript meets your expectations and sincerely welcome any further questions or suggestions.

With our sincere appreciation,

The authors.

Please see our detailed responses below in blue.

Reviewers' Comments:

Reviewer #1

Remarks to the Author:

Comments to the Authors:

In the manuscript, the authors investigated the link between excessive Era/histone lactylation and impaired endometrial receptivity in women with polycystic ovary syndrome (PCOS). These data provide new perspectives on the potential therapeutic strategy for endometrial dysfunction. This article represents an interesting contribution to the field, as the work presented is original, and the content is significant. Nevertheless, the article requires major revisions, as the current version is not suitable for publication. Several queries need to be addressed, and corrections are necessary. Providing additional details in the methods section, as well as supplementing and revising the results, would enhance the manuscript's value and comprehensiveness.

Response: We thank the reviewer for recognizing the potential interest of our work. We fully agree that the manuscript requires extensive revisions to improve the clarity, rigor, and completeness. We have thoroughly revised the manuscript in response to the constructive comments and believe that the overall quality of the work has been substantially enhanced.

Major concerns:

1-1. Abstract and Introduction

1. The Authors should explicitly state a research hypothesis. The introduction should conclude with the research hypothesis and the purpose of the study, rather than the obtained results.

Response: We appreciate the reviewer's insightful suggestion. In the revised manuscript, we have rephrased the final paragraph of the **Introduction** section to clearly articulate the research hypothesis and study objectives, rather than summarizing results:

“In this study, we hypothesized that women with PCOS are suffering from

endometrial dysfunction and consequent fertility decline, which might be the result of abnormal estrogen/progesterone signaling along with epigenetic alteration. To assess the endometrial impact on fertility in PCOS independent of embryonic factors, we conducted a retrospective cohort study ($n = 4,278$) to evaluate pregnancy outcomes between PCOS and non-PCOS controls, after correcting potential confounders via propensity score matching (PSM). We further employed primary endometrial samples of patients and the PCOS mouse model to confirm that ER α overexpression, instead of abnormal PGR expression, is responsible for impaired endometrial receptivity. We next selected histone lactylation as an example to study the interplay between the ER signaling and epigenetic modifications, given the fact that PCOS women were reported to have increased serum lactate levels³⁵. To evaluate whether the ER signaling affects histone lactylation, we generated *ESR1*-overexpressing (ESR1-OE) Ishikawa cells and applied MPP (an ER α inhibitor) to Ishikawa cells, confirming that ER α can regulate H3K181a (a major subtype of histone lactylation) via LDHA/LDHB. We also manipulated the lactate level in Ishikawa cells and the uterus of PCOS mice to determine the regulatory role of histone lactylation on ER α , demonstrating that inhibiting lactate production will decrease H3K181a, mitigate ER α overexpression, and recover impaired uterine receptivity. Together, our study lays a foundation for understanding endometrial dysfunction in PCOS and highlights the potential to improve the fertility of PCOS in the clinic by targeting histone lactylation.”

1-2. Materials and Methods

1. Repeating the experiment two or three times (2n or 3n, where n is the number of each experiment) may not be sufficient for a robust statistical analysis. Therefore, the authors should increase the number of experiments and explain the rationale for the observed variations in n across different experiments.

Response: We understand the reviewer’s concern about the statistical robustness. We agree with the reviewer that the increased sample size will enhance the statistical power. Accordingly, in the revised manuscript, we have increased the number of biological

replicates to at least five per group for most experiments, except for the CUT&Tag assay, which includes four biological replicates per group due to the larger sample volume required. It is not very practical to employ too large sample sizes in many situations, especially in clinical settings, due to the limitations of time, cost, and sample availability. Our sample sizes have been larger than or at least comparable to those in published papers¹⁻⁷. For CUT&Tag, the number of replicates used in our study also equals or exceeds the number reported in previous publications, where three or fewer replicates were commonly used^{8,9}. Regarding the varied n across different experiments, they were determined by the sample availability, including the difficulty in acquiring samples and sample volumes used in each experiment, the sensitivity/specificity of the experiment, as well as the success rate of the experiment.

To further confirm the robustness of our conclusions in the presence of clinical and biological heterogeneity, we employed following efforts: i) To minimize the impact of sample variation, we have carefully checked the normality of the data in each experiment to enable the correct application of the Student's *t*-test, and we would use the Mann-Whitney *U* rank-sum test instead when the data did not follow a normal distribution; ii) We would not make any conclusions based on a single experiment, and have validated our observations using multi-dimensional evidence, including phenotypes, gene expression, protein expression, and permutation experiments; iii) We would evaluate the consistency between replicates when necessary. In the CUT&Tag analysis, the H3K181a signal was consistent with each other among replicates from the same condition (**Supplementary Fig. 4k**):

Consistent H3K181a signals between biological replicates (n = 4 for each group).

We hope these clarifications adequately address the reviewer's concern and demonstrate the rigor of our experimental design.

2. What was the cell viability of the human primary endometrial epithelial and stromal cells?"

Response: The viability of human primary endometrial epithelial and stromal cells was evaluated using the 7-AAD exclusion assay. As shown in **Supplementary Fig. 3e**, 79.6% of epithelial cells and 89.2% of stromal cells were viable. To further validate stromal cell viability, we performed an in vitro proliferation assay. This does not apply to epithelial cells since they undergo terminal differentiation and cannot propagate in vitro. For stromal cells, proliferation dynamics following isolation and first-passage culture were monitored daily using the Incucyte® live-cell analysis system. Stromal cells continued to proliferate until day 7, when their growth reached a plateau phase, indicating sustained viability over the culture period (**Supplementary Fig. 3f**). These results are helpful to enhance the reliability of our conclusions, and we thus have added them to the revised manuscript (**Supplementary Fig. 3e-f**):

e, Flow cytometry plots showing 7-AAD-negative (viable) and 7-AAD-positive (non-viable) cell populations in primary epithelial and stromal cells. *f*, Longitudinal cell viability observation of primary endometrial stromal cells during in vitro culture for 7 days.

3. Detection of L-lactate content:

a) Were any controls included to validate the L-lactate detection assay?

Response: According to the manufacturer's protocol (Solarbio, bc2235), the Limit of Detection (LOD) of this assay is 0.03 $\mu\text{mol/mL}$, which is much smaller compared to the lowest value of our detection. We also used ddH₂O to establish the Limit of Blank (LOB), which is 0.01 $\mu\text{mol/mL}$ as the negative control, and the L-lactic acid (Standard) to develop the positive control, which is 10.00 $\mu\text{mol/mL}$ (**Reviewer Figure 1**). These observations confirmed the authenticity and reliability of the L-lactate detection assay.

Reviewer Figure 1. Lactate concentrations in negative and positive controls ($n = 5$).

b) What are the sensitivity and specificity of the assay used to measure L-lactate?

Response: For sensitivity, the LOD of the L-lactate measurement is 0.03 $\mu\text{mol/mL}$, much smaller than our detected values. For specificity, this assay is highly specific for L-lactate¹⁰⁻¹². According to the preliminary results of the manufacturer, neither D-lactate nor DL-lactate can be detected by this assay (**Reviewer Figure 2**):

Reviewer Figure 2. The L-lactic acid standard curve.

4. Immunostaining: Please provide information on negative controls, species reactivity of antibodies, and the type and brand of the fluorescence microscope used.

Response: Negative controls of all antibodies used in the immunostaining were included in the revised manuscript (**Supplementary Data Fig. 7**). Parallel tissue sections or cell samples were incubated with isotype control antibodies or processed without primary antibodies to exclude non-specific staining.

Supplementary Data Figure 7. Negative control images of antibodies used in the immunostaining. a, Representative images of immunohistochemistry staining along with negative controls (NC) in the human endometrium or murine uterus. b,

Representative images of immunofluorescence staining along with NC in the human endometrium, human primary endometrial epithelial cells and stromal cells, and murine uterus. Scale bar: 100 μ m.

Antibody specifications (including host species, catalog numbers, dilutions, and incubation conditions) are provided in the **Methods** section and **Supplementary Data**

Table 1:

Antibodies	Source	Identify	Dilution	Species Reactivity
Rabbit anti-IGFBP1 antibody	CST	31025T	1:1000	Human
Rabbit anti-Progesterone Receptor antibody	CST	8757S	1:1000	Human
Rabbit anti-Estrogen Receptor alpha antibody	Abcam	ab32063	1:1000	Mouse, Rat, Human
Rabbit anti-COX2 antibody	Abcam	ab179800	1:200	Mouse, Rat, Human
Goat anti-HOXA10 antibody	Abcam	ab191470	1:100	Human, Mouse, Rat, Rabbit, Horse, Chicken, Cow, Pig, Chimpanzee, Monkey, Gorilla
Rabbit anti-Ki67 antibody	Abcam	ab16667	1:200	Mouse, Rat, Human
Rabbit anti-MUC1antibody	Abcam	ab109185	1:200	Mouse, Rat, Human
Rabbit anti-Cytokeratin antibody	Abcam	ab52625	1:10000	Mouse, Human
Mouse anti-Vimentin antibody	Abcam	ab8978	1:1000	Mouse, Rat, Human, Pig, Zebrafish
Rabbit anti-pan K1a antibody	PTM BIO	PTM-1401	IF:1:50, WB:1:1000	All species
Rabbit anti-H3K181a antibody ChIP Grade	PTM BIO	PTM-1427RM	Cut&tag:1:50	Human, Mouse
Rabbit anti-H3K181a antibody	PTM BIO	PTM1406RM	1:1000	Human, Mouse, Rat
Rabbit anti-H4K121a antibody	PTM BIO	PTM1411RM	1:1000	Human, Mouse, Rat
Rabbit anti-Histone H4 antibody	PTM BIO	PTM1015RM	1:2000	Human, Mouse, Rat, Monkey, Pig, S.cerevisiae
Rabbit anti-Histone H3 antibody	PTM BIO	PTM1002RM	1:2500	Human, Mouse, Rat, Rice
Mouse anti-ER α Antibody	Immunoway	YM0252	1:200	Human
Rabbit anti-LDHB antibody	Proteintech	14824-1-AP	1:10000	Human, Mouse, Rat, Pig, Chicken, Bovine, Sheep
Rabbit anti-LDHA antibody	Proteintech	19987-1-AP	1:5000	Human, Mouse, Rat, Rabbit, Chicken, Goat
Rabbit anti-GAPDH antibody	Abcam	ab181602	1:10000	Mouse, Rat, Chicken, Human, Zebrafish, African green monkey, Xenopus tropicalis
Mouse anti-beta ACTIN antibody	Abcam	ab6276	1:10000	Mouse, Rat, Cow, Dog, Human, African green monkey, Chinese hamster
HRP-conjugated Affinipure Donkey anti-Goat IgG(H+L)	Beyotime	A0181	1:1000	Unspecified reactive species
HRP-conjugated Affinipure Goat Anti-Rabbit IgG(H+L)	Beyotime	A0208	1:1000	Unspecified reactive species
HRP-conjugated Affinipure Goat Anti-Mouse IgG(H+L)	Beyotime	A0216	1:1000	Unspecified reactive species
Phalloidin-iFluor 555 Reagent	Abcam	ab176756	1:200	Unspecified reactive species
Phalloidin-iFluor 488 Reagent	Abcam	ab176753	1:200	Unspecified reactive species

Imaging was conducted using a Zeiss LSM 880 confocal microscope with Airyscan super-resolution capability, which is now indicated in the **Methods** section:

“Images were captured using a Zeiss LSM 880 confocal microscope with Airyscan super-resolution capability.”

5. Plasmids and cell transfection:

a) *Were any specific conditions or treatments applied to improve transfection efficiency?*

Response: We thank the reviewer for this important question. To optimize the transfection efficiency, we have adopted a series of strategies, which are indicated in

the **Methods** section in the revised manuscript (revised text in bold):

“**Ishikawa cells were seeded in 6-well plates at a concentration of 2×10^5 cells per well and cultured to a confluence of approximately 60%, then transfected with the lentiviral control virus HBLV-ZsGreen-PURO or with the target virus HBLV-ESR1-3XFLAG-ZsGreen-PURO (Hanbio Biotechnology) using Lipofectamine 3000 (Invitrogen) according to the manufacturer’s instructions.** After 48 hours of infection, cells that were not infected efficiently were killed by adding and maintaining 1.0 $\mu\text{g/mL}$ puromycin. In this way, a stable cell line with ESR1 **overexpression**, maintained under puromycin selection, was finally obtained.”

b) *What was the cell viability post-transfection?*

Response: Cell viability was evaluated using the 7-AAD exclusion assay. We observed 95.9% and 99.9% of cells remain viable before and after transfection (**Supplementary Fig. 5b**), confirming that the transfection did not induce cytotoxicity. Furthermore, cell proliferation was monitored for 7 days using the Incucyte® live-cell imaging system. Both control and ESR1-OE cells exhibited similar growth kinetics, reaching a plateau phase by day 7, indicative of sustained cell viability (**Supplementary Fig. 5c**). We included these results in the revised manuscript:

b, Flow cytometry plots showing 7-AAD-negative (viable) and 7-AAD-positive (non-viable) cell populations in pre- and post-transfected Ishikawa cells. c, Longitudinal cell viability observation in CON-OE and ESR1-OE Ishikawa cells during in vitro culture for 7 days.

6. *Western blot: Please provide details on negative controls, species reactivity of the antibodies, and the method used for analysing the results.*

Response: We thank the reviewer for the suggestion. We loaded 1 × loading buffer without protein samples as the negative control for each WB experiment, ensuring the specificity of antibodies (**Supplementary Data Fig. 6**):

Supplementary Data Figure 6. Full-length images of western blotting membranes.

Antibody specifications are provided in the **Methods** section and **Supplementary Data Table 1**. See our response to **Comments 1-2-4** for details. The method used for analyzing WB results is provided in the **Methods** section (revised text in bold):

“Specific primary and secondary antibodies were then incubated with membranes sequentially. **The blots were visualized using an enhanced chemiluminescent substrate (Thermo Fisher Scientific) and quantified with ImageJ software, with normalization to GAPDH, ACTIN, H3, or H4. Details of antibodies used for western blotting were summarized in Supplementary Data Table 1.**”

7. Quantitative Real-Time PCR (qRT-PCR):

a) Please adhere to MIQE guidelines.

Response: Thanks for the important comment, and we have conducted qRT-PCR following MIQE guidelines rigorously¹³:

i) The primer specificity was validated by agarose electrophoresis. After PCR amplification with the used primers, only a single clear band was detected, and the band size is consistent with the amplification products as expected (**Reviewer Figure 3**):

Reviewer Figure 3. Electrophoresis validation of the primer specificity.

The size of amplification products of primers is listed in **Supplementary Data Table**

2:

Gene	Forward (5' to 3')	Reverse (3' to 5')	GenBank accession numbers	Amplicon Size	R2	Amplification efficiency	Slope	Primer Concentrations
human-GAPDH	GGAGCGAGATCCCTCCAAAAT	GGCTGTTGCATACTTCTCATGG	AK299972.1	197bp	0.998	96.048	-3.420	10 μ M
human-18S	GTAACCCGTTGAACCCGATT	CCATCCAATCGGTAGTAGCG	8UJ9_S2	151bp	0.999	94.435	-3.463	10 μ M
human-ACTB	TGCCCATCTACGAGGGGTAT	CTTAATGTACGCACGATTTC	PQ040393.1	152bp	0.999	94.675	-3.457	10 μ M
human-EP300	GCTTCAGACAAGTCTGGCAT	ACTACCAGATCGCAGCAATTC	BC053889.1	79bp	0.993	103.720	-3.236	10 μ M
human-LDHA	TTGACCTACGTGGCTTGAAG	GGTAACGGAATCGGGCTGAAT	BC051361.1	91bp	0.993	106.879	-3.167	10 μ M
human-LDHB	TCTGTGACCGCAATTCTAAGA	GCACCAGATTGAGCCGACTC	EU919185.1	82bp	0.993	103.949	-3.231	10 μ M
human-SIRT1	AAGTTGACTGTGAAGCTGTACG	TGCTACTGGTCTTACTTTGAGGG	JQ768366.1	218 bp	0.996	96.389	-3.412	10 μ M
human-SIRT2	ATCCACCGCCTCTATGACAA	CGCATGAAGTAGTGACAGATGG	KF032391.1	162bp	0.997	109.280	-3.118	10 μ M
human-SIRT3	CCCAAGCCCTTTTCACTTT	CGACACTCTCTCAAGCCCA	AF083108.2	148bp	0.995	91.038	-3.557	10 μ M
human-PRL	ATCATCTGGTCACGGAAGTACG	GGTTTGCTCCTCAATCTACAG	BC088370.1	83bp	0.990	94.216	-3.469	10 μ M
human-IGFBP1	TTGGGACGCCACTCAGTACCTA	TTGGCTAAACTCTACGACTCT	FJ795026.1	114bp	0.995	109.335	-3.117	10 μ M
human-ESR1	CCCCTCAACAGCGTGTCTC	CGTCGATTATCTGAATTTGGCCT	LC516420.1	180bp	0.999	102.598	-3.261	10 μ M
human-MSX1	ACACAAGACGAACCGTAAGCC	CACATGGGCCGTGTAGAGTC	HM213930.1	382bp	0.993	96.266	-3.416	10 μ M
human-MSX2	TGCAGAGCGTGCAGAGTTC	GGCAGCATAGGTTTTGCAGC	AH004951.2	144bp	0.997	98.052	-3.369	10 μ M
human-LTF	CCCAGGAACCGTACTTCAGC	GTCACACAACGGCATGAGA	KT006756.1	219bp	0.992	104.852	-3.211	10 μ M
human-PGR	CCCAGCATGTGCGCTTAGAAA	AGTGCTCTCACAACTCGACTT	AY382151.1	96bp	0.999	109.888	-3.106	10 μ M
human-AREG	GTGGTGCTGTGCTCTTGATA	CCCCAGAAAATGGTTCACGCT	BT019866.1	97bp	0.994	108.596	-3.132	10 μ M
human-IHH	TCCGTCAAGTCCGAGCACT	GTCCGTGAGTCTCGATGACCTG	BC136588.1	228bp	0.999	90.031	-3.586	10 μ M
mouse-Gapdh	TGGCCTCCGTGTTCCCTAC	GAGTTGCTGTTGAAGTCGCA	OX390160.1	178bp	0.990	105.802	-3.190	10 μ M
mouse-Actb	GTGACGTTGACATCCGTAAGA	GCCGGACTCATCGTACTCC	BC138614.1	245bp	0.997	109.838	-3.107	10 μ M
mouse-Tbp	ACCGTGAATCTTGGCTGTAAC	GCAGCAAATCGCTTGGGATTA	BC016476.1	86bp	0.992	109.964	-3.104	10 μ M
mouse-36b4	CTCACTGAGATTCGGGATATG	CTCCCACCTTGTCTCCAGTC	BC011291.1	223bp	0.993	103.523	-3.240	10 μ M
mouse-Esr1	TCTGCAAGGAGACTCGCTACT	GGTGCAATGGTTGTAGCTGGAC	LC260511.1	153bp	0.996	104.478	-3.219	10 μ M
mouse-Msx1	GCACAAGACCAACCGCAAG	CGCTCGGCAATAGACAGGT	BC016426.1	102bp	0.994	106.096	-3.184	10 μ M
mouse-Msx2	CTAAAGGCCTGACTTGTTCG	CGGCTTCTGTGCGGACATGAG	BC141132.1	161bp	0.990	102.222	-3.270	10 μ M
mouse-Ltf	GTCTGCCATTGGCTTTGTGAGG	CCTTTGAGGCTATCACATCTGC	BC006904.2	122bp	0.992	108.303	-3.138	10 μ M

ii) As suggested by the reviewer, we have implemented dual internal reference genes (*GAPDH* and *ACTIN* for humans, *Tbp* and *Gapdh* for mice) to ensure the stability of our qRT-PCR results.

iii) We have verified the amplification efficiency of primers, which was between 90% and 110% ($R^2 > 0.99$, standard curve method), with the slope of the qRT-PCR standard curve ranging from -3.6 to -3.1 (**Supplementary Data Figs. 1-4**), confirming the accuracy of using delta-delta CT for gene expression calculation.

iv) Negative controls were included in each qRT-PCR experiment.

v) The biological replicates of all qRT-PCR experiments are indicated, and the number of biological replicates in each group is no less than 5 to ensure the reliability.

b) A validation experiment is necessary to confirm the accuracy of the delta-delta CT calculation. What were the absolute values of the slope of delta CT vs. log input?

Response: According to MIQE guidelines, the primer amplification efficiency must fall within 90%-110% with $R^2 > 0.99$ to ensure the accuracy and reproducibility of qRT-PCR. The amplification efficiencies and absolute values of the slope of delta CT

were determined through standard curve analysis for all used primers. The results showed that amplification efficiencies of primers are in the optimal range, as well as the slope of delta CT (from -3.6 to -3.1), confirming the accuracy of the delta-delta CT calculation. These results were added to the revised manuscript as **Supplementary Data Figs. 3-4:**

Supplementary Data Figure 3. The amplification efficiency of human target genes.

Supplementary Data Figure 4. The amplification efficiency of mouse target genes.

c) Using a single reference gene for qRT-PCR analysis is insufficient. Additionally, the stability of the reference gene in this study should be assessed.

Response: We agree with the reviewer that an extra reference gene for qRT-PCR analysis is necessary. We have also carefully assessed the stability of the reference gene as suggested, using NormFinder according to MIQE guidelines. Conventional reference genes, including *GAPDH*, *ACTB* (β -ACTIN), *RNA18S1* (18S rRNA), *Tbp*, and *Rplp0* (36b4), were evaluated for expression stability using geNorm algorithms^{14,15}. In human samples, *GAPDH* (Stability value = 0.257), *ACTB* (Stability value = 0.752), and 18S rRNA (Stability value = 0.881) were analyzed, with *GAPDH* and *ACTB* selected as optimal reference genes (**Reviewer Figure 4a**). For the mouse, *Tbp* (Stability value = 0.004) and *Gapdh* (Stability value = 0.004) demonstrated superior stability compared to *Actb* (Stability value = 0.011) and *36b4* (Stability value = 0.014) (**Reviewer Figure 4b**).

Reviewer Figure 4. Stability analysis of housekeeping genes. a, Amplification curves, melt curves, and stability values of human housekeeping genes. b, Amplification curves, melt curves, and stability values of mouse housekeeping genes.

In addition, we also confirmed that housekeeping gene primers demonstrate optimal amplification efficiency (90%-110%) and specificity, as evidenced by standard curve validation ($R^2 > 0.99$) and single-peak melt curves (**Supplementary Data Figs. 1-2**):

Supplementary Data Figure 1. The amplification efficiency of human reference genes.

Supplementary Data Figure 2. The amplification efficiency of mouse reference genes.

Gene expression normalized against different reference genes showed high concordance across different experimental conditions (**Reviewer Figure 5**). We thus selected *GAPDH* as the human reference gene and *Tbp* as the mouse reference gene to present qRT-PCR results in the revised manuscript. We would like to include all the normalization results in the manuscript if the reviewer thinks it is necessary.

Reviewer Figure 5. Relative mRNA levels normalized to two reference genes. a, Relative mRNA levels of *ESR1*, *MSX1*, *MSX2*, and *LTF* normalized to *GAPDH* and *ACTB* in the human mid-secretory endometrium ($n = 25$ for CON and $n = 12$ for PCOS group). **b,** Relative mRNA levels of *PGR*, *AREG*, and *IHH* normalized to *GAPDH* and *ACTB* in the human mid-secretory endometrium ($n = 25$ for CON and $n = 12$ for PCOS group). **c,** Relative mRNA levels of *LDHA* and *LDHB* normalized to *GAPDH* and *ACTB* in Ishikawa cells between the CON-OE and ESR1-OE groups ($n = 5$ for each group).

d, Relative mRNA levels of EP300, SIRT1, SIRT2, and SIRT3 normalized to GAPDH and ACTB in Ishikawa cells between the CON-OE and ESRI-OE groups (n = 5 for each group). e, Relative mRNA levels of IGFBP1 and PRL normalized to GAPDH and ACTB in stromal cells during induced decidualization (n =6 for each group). f, Relative mRNA levels of Esr1, Msx1, Msx2, and Ltf normalized to Tbp and Gapdh in the murine uterus on day 4 (n = 6 for each group).

d) What negative controls were included in qRT-PCR? Were the primers designed to overlap exon-exon boundaries? How was potential DNA contamination controlled?

Response: Ultrapure water was included as a negative control in all qRT-PCR experiments, with no detectable amplification observed (**Supplementary Data Fig. 5**):

Supplementary Data Figure 5. Negative controls of qRT-PCR.

Primers were designed to span exon-exon boundaries (verified by Primer-BLAST). To control DNA contamination, we used DNase-treated RNA and employed sterile techniques including filtered pipette tips and a DNA-free tube. In addition, the reverse transcription reagent we used can complete the genomic DNA clearance and reverse

transcription reaction, avoiding contamination from genomic DNA and eliminating the need for a separate RT control mix.

e) Providing a detailed description of qRT-PCR conditions in a table format would improve clarity. This should include primer sequences, primer concentrations, PCR amplicon lengths and GenBank accession numbers.

Response: We thank the reviewer for this suggestion. We have added the detailed description of qRT-PCR conditions to the revised manuscript as the **Supplementary Data Table 2**. See our response to **Comments 1-2-7-a** for details.

8. *CUT&Tag assay:*

a) What was the starting number of cells before nuclei isolation, and how many nuclei were obtained per sample?

Response: Following Trypan Blue viability assessment (Solarbio, C0040-100ml), each biological replicate underwent nuclei isolation from approximately 5×10^5 input cells, yielding $2.1 \pm 0.3 \times 10^5$ intact nuclei ($22.4 \pm 3.1\%$ recovery) for CUT&Tag analyses, which is acceptable according to the manufacturer's instructions¹⁶.

b) How did you assess the effectiveness of the antibodies used?

Response: For CUT&Tag, we employed a well-characterized commercial H3K18la antibody (PTM BIO, PTM-1427RM), which has been extensively validated in prior studies¹⁷⁻¹⁹. At an optimized dilution of 1:50, this antibody consistently generates high-quality libraries. Also, this antibody is highly effective in our immunostaining experiments.

c) What negative and positive controls were included in the analysis?

Response: We thank the reviewer for this question. CUT&Tag typically does not require INPUT as ChIP-seq since it directly uses Tn5 transposase-cleaved DNA fragments for analysis, avoiding the complex process of background noise processing¹⁶. For negative controls, we employed isotype-matched IgG antibodies obtained from the

same manufacturer, host species, and purification process as the H3K18la antibody. The IgG negative control did not exhibit specific detectable enrichment and demonstrated low library output (**Reviewer Figure 6a-b**). We utilized published H3K18la CUT&Tag data and known sites with H3K18la enrichment as positive controls²⁰. Our samples demonstrate significant enrichment at both the *TTK* and *BUB1B* gene loci (**Reviewer Figure 6c**).

Reviewer Figure 6. Quality control of the H3K18ac CUT&Tag assay. a, Quality assessment of IgG and H3K18la CUT&Tag library construction in both CON and PCOS groups. b, The H3K18la signal around gene bodies in the human mid-secretory endometrium. c, The H3K18la signal at selected positive control genes.

1-3. Results, Figures and Conclusions

1. To properly assess decidualization, protein levels of decidual markers, such as PRL and IGFBP1, should be measured, rather than relying solely on their gene expression levels.

Response: As suggested by the reviewer, we have quantified protein levels of IGFBP1 and PRL (Fig. 1c-e; Supplementary Fig. 3h). As a secreted protein, the level of PRL is typically measured using ELISA kits²¹⁻²³. We thus quantified extracellular PRL concentrations in cell culture supernatants using a commercial ELISA kit (H095-1-2, Jiancheng Bio). Notably, both IGFBP1 and PRL levels were significantly reduced in PCOS during decidualization compared to controls:

c, Representative images illustrating protein levels of IGFBP1 in human proliferative endometrial stromal cells during induced decidualization ($n = 5$ for each group). d, ELISA analysis of PRL protein levels in cell culture supernatants from primary stromal cells during induced decidualization ($n = 6$ for each group). e, Relative mRNA levels of IGFBP1 and PRL in stromal cells during induced decidualization ($n = 6$ for each group).

h, Relative protein levels of IGFBP1 in human endometrial stromal cells during induced decidualization. ($n = 5$ for each group).

2. Immunostaining: Figures should include images of negative controls.

Response: We thank the reviewer for this comment. We have now included images of negative controls in the revised manuscript as **Supplementary Data Fig. 7**. See our response to **Comments 1-2-4** for details.

3. Immunostaining analysis primarily serves a qualitative purpose, confirming protein presence and localisation. The conclusions drawn from these results should not be based solely on the expression levels inferred from immunostaining.

Response: We understand the reviewer's concern and agree with the reviewer that the immunostaining analysis is not reliable as a quantification method. In our initial submission, we assessed the expression levels of proteins including ER α , PGR, COX2, HOXA10, Pan Kla, H3K18la, and H4K12la using immunostaining for the human endometrium. Except for COX2 and HOXA10, all proteins were also validated by WB. We have now supplemented WB results for COX2 and HOXA10 in the revised manuscript (**Fig. 1h-i**):

h and i, Protein levels of COX2 and HOXA10 in the human mid-secretory endometrium ($n = 10$ for each group).

For immunohistochemistry (IHC) staining of Ki-67, a marker of cell proliferation, we counted the number of positive cells to clarify the proliferation status of epithelial and stromal cells. And we will be aware of the limitations of such analysis when making conclusions.

In mouse samples, ER α , PGR, and H3K18la were detected by both immunostaining and WB. However, MUC1, which is primarily expressed in endometrial epithelial cells, could not be reliably detected by WB due to its low expression. Similarly, we will avoid concluding any quantitative results solely from this experiment, and Ki-67 IHC staining

was performed together to evaluate the proliferation status.

4. *Western blot figures: Adding a molecular weight marker to the presented blots would facilitate the interpretation of the protein size and improve result analysis.*

Response: We are grateful for the suggestion and have added the weight marker for all WB figures in the revised manuscript. For instance (**Fig. 3a**):

Protein levels of ERα and H3K18la in Ishikawa cells (n = 6 for each group).

5. *Concerns regarding the reference protein panels in Figure 1C&I (GAPDH), Figure 2E (H3), and Figure 2G (H4): The intensity of these proteins differs between control and PCOS groups. Reference proteins should not show significant variations between samples. Given this discrepancy, statistical analyses should be conducted on these proteins, and the results should be presented.*

Response: We understand the reviewer's concern and agree that reference proteins should not show significant variations between samples. Following optimization of protein concentrations, we repeated both reference and target protein assays. Quantitative analysis was also provided to confirm the robustness of our observations: **Fig. 1c, l, and m**:

c, Representative images illustrating protein levels of IGFBP1 in human proliferative endometrial stromal cells during induced decidualization ($n = 5$ for each group). *l-m*, Protein levels of ER α and PGR in the human mid-secretory endometrium ($n = 10$ for each group).

Fig. 2d-g:

d-g, Protein levels of H3K18la and H4K12la in the human mid-secretory endometrium ($n = 5$ for each group).

6. In Figure 2, the text states that experiments included at least three biological replicates, whereas the CUT&Tag methodology description mentions only two replicates. This discrepancy should be clarified.

Response: We have expanded our sample size to 4 biological replicates per group and revised the figure legends to ensure a clear clarification (revised text in bold):

“Heatmaps of the H3K18la signal around gene bodies in the human mid-secretory endometrium ($n = 4$ for each group).”

7. Each figure should be self-explanatory. Abbreviations in figure legends should be defined to ensure that figures can be interpreted independently of the main text.

Response: We thank the reviewer for this suggestion. We systematically reorganized all figure legends to optimize clarity and accuracy, ensuring each panel: i) presents data without ambiguity; ii) includes necessary scale bars/annotations; iii) indicates sample sources, groups, and the number of biological replicates clearly; iv) explains abbreviations in each figure legend.

Reviewer #2

Remarks to the Author:

In this manuscript, Shan et al. attempt to define the mechanisms underlying impaired endometrial receptivity in women with PCOS. To answer this question, they use correlative studies in humans and functional tests in mice. The main findings of the study are:

1. Strictly controlling for embryonic factors and potential confounders, a large cohort study shows a lower implantation rate in women with PCOS. This suggests that endometrial abnormalities contribute to the reproductive problems in PCOS and are a major factor in poor pregnancy outcomes.

2. The authors show increased levels of ER α and H3K181a in the mid-secretory endometrium of women with PCOS and also demonstrate that overexpression of ER α increases H3K181a via LDHA/LDHB in Ishikawa cells.

3. Impaired embryo implantation is still observed when blastocysts are transferred from control mice to PCOS mice, suggesting endometrial dysfunction as a possible cause of reproductive failure in PCOS. This defect could be significantly improved by reducing H3K181a in the uterus of PCOS mice, suggesting a potential therapeutic target to improve reproductive outcomes in PCOS.

Overall, the proposed regulatory model is well supported by the mouse data, and the human data provide valuable insights into the mechanisms of endometrial dysfunction in PCOS. The manuscript is well written and well organized. Most of the conclusions are robust and consistent with the results. This work will be of great interest to a wide audience in the field. However, the regulatory relationship between H3K181a and ER α needs further clarification and several important questions need to be addressed before publication in Nature Communications.

Response: We thank the reviewer for recognizing the potential interest of our work and have conducted extensive edits and analyses to address these concerns.

2-1 For the retrospective cohort study:

a) The selection of covariates is a crucial step in PSM as it directly affects the

effectiveness of the matching process and the reliability of the subsequent analysis. Please describe the steps and principles that the authors followed in selecting the variables.

Response: This information was added to the **Methods** section in the revised manuscript:

“The selection of covariates for PSM was guided by the following principles: we prioritized variables consistently reported as confounders or predictors of reproductive outcomes in PCOS and infertility studies, including female age, BMI, infertility duration, miscarriage history, and male age, all of which are associated with pregnancy success or live birth rates⁶²⁻⁶⁶. We excluded variables lying on causal pathways between PCOS (exposure) and reproductive outcomes (e.g., AMH, AFC) to prevent over-adjustment.”

b) Elevated BMI is a typical feature of PCOS. Would the inclusion of BMI in the PSM lead to over-matching?

Response: While elevated BMI is a hallmark of PCOS in European countries, there is no significant difference in BMI between PCOS and non-PCOS in China²⁴. Also, BMI may independently affect reproductive outcomes through metabolic pathways²⁵. Excluding BMI would leave residual confounding, as its effect on outcomes is not fully mediated by PCOS diagnosis. Over-matching occurs when a variable is both a consequence of exposure and unrelated to the outcome. However, BMI is a pre-exposure confounder (before PCOS diagnosis) and strongly linked to outcomes. We thus concluded that including BMI in the PSM is unlikely to lead to over-matching.

c) Exclusion criteria for the cohort study include endometriosis and RSA. The authors should clarify the diagnostic criteria for endometriosis and RSA.

Response: We appreciate the reviewer’s suggestion. We have rephrased the **Methods** section to include such necessary information:

“Endometriosis is defined as a chronic inflammatory condition characterized by the presence of endometrial-like tissue (glands and stroma) outside the uterine cavity,

primarily affecting pelvic organs (e.g., ovaries, peritoneum), often leading to pain and/or infertility⁶⁷. Recurrent implantation failure (RIF) is defined as the failure to achieve a clinical pregnancy after at least three transfer cycles involving ≥ 4 high-quality embryos⁶⁸. Recurrent spontaneous abortion (RSA) refers to two or more consecutive pregnancy losses before 28 weeks of gestation with the same partner, including biochemical pregnancies.”

d) It is mentioned that endometrial tissue will be collected from different menstrual cycles. How are the different menstrual cycles determined?

Response: We have added this information to the **Methods** section:

“Endometrial tissues were collected at different stages of the menstrual cycle, which was determined by the last menstrual period, serum LH peak levels, and confirmation of ovulation using transvaginal ultrasound. The proliferative phase, menstrual cycle days 5-14; the early secretory phase, 1-4 days post-ovulation; the mid-secretory phase, 5-8 days post-ovulation; the late secretory phase, 9-14 days post-ovulation. The early pregnancy status was confirmed by ultrasonographic visualization of an intrauterine gestational sac. The phase of endometrial samples was further confirmed by two senior pathologists using examination of Hematoxylin and Eosin (H&E) stained sections.”

2-2 For analysis using the human endometrium:

a. In addition to PGR and HOXA10, were changes in HAND2 expression or activity observed?

Response: As a key mediator of the progesterone receptor (PGR) signaling in endometrial cells, HAND2 is mainly expressed in endometrial stromal cells, regulated by PGR, and mediates the inhibitory effect of progesterone on the proliferative effect of estrogen²⁶⁻²⁸. In this study, PCOS mid-secretory endometrium was characterized by abnormally activated estrogen signaling, while no significant differences were observed in PGR protein levels and progesterone-responsive gene expression. Thus, the protein levels of HAND2 were also not significantly different between CON and PCOS groups

(Reviewer Figure 7).

Reviewer Figure 7. Protein levels of HAND2 in the human mid-secretory endometrium (n = 10 for each group).

b. At what time points were the samples in Fig. 1c collected?

Response: In **Fig. 1c**, endometrial samples were isolated from the CON and PCOS groups in the proliferative phase. This information was added to the figure legend of **Fig. 1c** in the revised manuscript (revised text in bold):

“Representative images illustrating protein levels of IGFBP1 in human proliferative endometrial stromal cells during induced decidualization (n = 5 for each group).”

c. Lines 118-120: Please include representative images of untreated cells at day 0 to illustrate the morphological changes between the experimental groups.

Response: We have added the representative images of untreated cells on day 0 to the revised manuscript (**Fig. 1b**):

Immunofluorescence (IF) staining of cell morphologies in human endometrial stromal cells during induced decidualization. Decidualization was induced by the treatment

with 0.5 μ M cAMP and 1 μ M MPA. The F-actin cytoskeleton was visualized by rhodamine phalloidin staining. Scale bar: 100 μ m.

d. Line 127: The statement implies that LIF is a downstream gene of PGR. Is there direct evidence to support this? Please clarify this.

Response: Thank you for this important comment. LIF, a cytokine highly expressed in the endometrial glandular epithelium on the fourth day of pregnancy, can participate in the establishment of endometrial receptivity by activating the downstream STAT3 signaling pathway, considered to be one of the markers of endometrial receptivity^{29,30}. However, by reviewing the literature and the ChIP-seq data of PGR, we found that there is still a lack of direct evidence on PGR regulating LIF gene expression. Therefore, we revised the manuscript to avoid this imprecise statement.

e. Fig. 2A-B, I-J: The scale bars in these panels appear to be inconsistent in length, although they are indicated as 100 μ m. The authors should check the accuracy of the scale bars.

Response: We thank the reviewer for this comment. We have re-examined all scale bars in **Fig. 2** and replaced all scale bars with standardized 100 μ m indicators.

a, H3K18la and H4K12la immunofluorescence (IF) staining in the endometrium of different hormonal stages. h-i, H3K18la and H4K12la IF staining in the human mid-

secretory endometrium. Scale bar: 100 μ m.

d. Please explain the method used to normalize the CUT&Tag data.

Response: The CUT&Tag data was normalized using the RPGC method, which adjusted read counts per region based on sequencing depth, ensuring comparability across samples regardless of sequencing depth. We added this information to the revised manuscript in the **Methods** section:

“Normalization was performed with the RPGC method.”

e. Does the inhibition of ER α have an effect on LDHA/LDHB?

Response: Inhibiting ER α in Ishikawa cells significantly downregulated LDHA and LDHB (**Reviewer Figure 8**), further confirming the regulatory effect of ER α on histone lactylation:

Reviewer Figure 8. ER α inhibition affects the LDHA/LDHB expression and H3K18la level. a, Immunofluorescence staining of ER α and H3K18la in Ishikawa cells between the CON and MPP-treated (5 μ M) groups. b, Intracellular lactate levels in

Ishikawa cells (n = 6). c-d, Protein levels of ER α and H3K18la in Ishikawa cells (n = 6). e-f, Protein levels of LDHA and LDHB in Ishikawa cells (n = 6). Scale bar: 100 μ m.

h. In addition to cell line experiments, it would be very valuable to include organoids from patient samples. This would provide more physiologically relevant evidence and further elucidate the underlying mechanisms.

Response: We thank the reviewer for this question and agree that organoids would provide more physiologically relevant evidence. We identified a marked upregulation of histone lactylation (Pan Kla, H4K12la, H3K18la) in PCOS patient-derived endometrial organoids, confirming the excessive endometrial histone lactylation in PCOS (**Supplementary Fig. 4h-j**):

h-j, Pan Kla, H3K18la, and H4K12la IF staining in human endometrial organoids. Scale bar: 100 μ m.

2-3 For the mouse model:

a. LINE179-180: The authors should briefly discuss the differences in endometrial physiology between humans and mice.

Response: An important difference between human and mouse endometrial physiology is the decidualization of stromal cells. Decidualization of the human endometrium involves a dramatic morphological and functional differentiation of human endometrial stromal cells. The human decidua is formed routinely and is shed off in the absence of an embryo in the endometrium³¹. However, in mice, decidualization of stromal cells occurs after successful embryo implantation³². We highlighted this important difference in the **Discussion** section in the revised manuscript:

“For instance, the human decidua is formed routinely and is shed off in the absence

of an embryo, while decidualization of stromal cells occurs after successful embryo implantation in mice⁵⁵.”

b. In Fig. 4, the scale bar is missing in several figures. Please add the scale bar to ensure accurate interpretation of the images.

Response: We sincerely appreciate the reviewer’s meticulous examination of our manuscript. We have added standardized scale bars to all relevant figures.

c. The authors should investigate the functions of stromal cells in PCOS and treated mice, e.g. apoptosis.

Response: We thank the reviewer for this suggestion. We have now included TUNEL staining results in the revised manuscript, demonstrating the elevated apoptosis level in PCOS mice (**Supplementary Fig. 6k**):

The TUNEL assay was used to detect apoptosis in the murine uterus on day 4. Scale bar: 100 μ m.

d. Endometrial lactate levels are elevated in PCOS patients, which has not been studied in PCOS mice. This is important to confirm the robustness of the finding in different species.

Response: We thank the reviewer for this suggestion. Our analyses revealed that PCOS mice exhibited significantly elevated uterine lactate levels compared to controls, consistent with our findings in the human endometrium (**Supplementary Fig. 7g**):

Lactate levels in the murine uterus on day 4 (n = 7 for each group).

e. While lactate is essential for normal embryo implantation, excessive H3K18la is detrimental in PCOS. The authors should discuss the importance of maintaining adequate intrauterine lactate levels for successful embryo implantation.

Response: Thanks for the valuable suggestion. We have highlighted this important point in the **Discussion** section in the revised manuscript (revised text in bold):

“**Lactylation, a lactate-induced histone modification,** plays a significant role in reproductive health, including gametogenesis, embryogenesis, endometrial decidualization, and placentation^{38,59-61}. **It has been reported that murine uterine decidual cells exhibit increased histone lactylation during mid-pregnancy, while inhibition of lactylation results in impaired uterine decidualization and pregnancy failure³⁰. However, excessive endometrial histone lactylation is also harmful.** Along with excessive ER α , we also observed elevated H3K18la in the endometrium of PCOS patients and the uterus of PCOS mice. ER α overexpression increased H3K18la via upregulating LDHA/LDHB in Ishikawa cells, and H3K18la, in turn, promoted ER α expression upon its increase. Notably, inhibiting **ER α or lactate production restored uterine** receptivity in PCOS mice, with improved implantation rates. **Thus, the histone lactylation level needs to be maintained properly to ensure endometrial receptivity and implantation.**”

f. In PCOS mice, it has not been tested whether inhibition of ER α could restore implantation and reduce H3K18la levels. This is important to confirm the effects of ER α on H3K18la.

Response: As suggested, we administered the ER α inhibitor MPP into the uterine horns of DHEA-treated mice on day 3. By day 5, MPP-injected DHEA mice exhibited a significant increase in implantation sites compared to the DHEA group (Supplementary Fig. 7a-c), featuring restored uterine branching and H3K181a level (Supplementary Fig. 7d-f):

a, Schematic diagram of PCOS model mice construction and MPP intervention. *b*, Representative images of implantation sites on day 5 in Oil, DHEA, and DHEA+MPP groups (8.14 $\mu\text{g}/\text{kg}$ MPP treatment). *c*, Number of implantation sites on day 5 in Oil, DHEA, and DHEA+MPP groups ($n = 8$ for each group). *d*, Hematoxylin and eosin (H&E) staining of uterine morphology on day 4 in Oil, DHEA, and DHEA+MPP groups. *e-f*, Uterine H3K181a levels on day 4 in Oil ($n = 6$), DHEA ($n = 7$), and DHEA+MPP ($n = 7$) groups.

Reviewer #3

Remarks to the Author:

Response: We thank the reviewer for co-reviewing our work.

Reviewer #4

Remarks to the Author:

Shan et al., have assessed the role of androgen (DHEA) exposure on endometrial lactulation-driven histone modifications that would also explain the higher ESRa expression and proliferation in PCOS endometrium. Blocking lactulation rescued the endometrium from excess estrogen effect.

The idea is interesting and approach is novel, however there are some concerns in the manuscript. The human data is also intriguing with appropriate statistical analysis.

Response: We sincerely thank the reviewer for recognizing the novelty and potential interest of our study. In response to raised concerns, we have carefully revised the manuscript and addressed each point in detail. We believe that these revisions have substantially improved the quality and clarity of the work.

4-1. The major concern is the poor quality of data and inadequately explained processes and samples.

Response: We understand the reviewer's concern. We also agree with the reviewer that the data quality needs to be improved and the processes and samples need to be explained in adequate detail. We have increased sample sizes to enhance the robustness of our conclusions. We employed at least five biological replicates for most experiments, except for CUT&Tag, which requires a larger sample volume. Nonetheless, four biological replicates in CUT&Tag are still larger or at least comparable to published studies. We have enhanced the data quality by rigorously following the MIQE guidelines in qRT-PCR experiments, including extra reference genes in qRT-PCR, maintaining stable levels of reference proteins in WB, presenting summarized statistical results, and implementing proper positive/negative controls for experiments. To improve the clarity of our manuscript, we have added extensive information, including a detailed description of sample collection, model establishment, inclusion/exclusion criteria of the cohort study, sample sizes, and agent specifications. We believe our extensive edits of the manuscript will increase the clarity and robustness of our

manuscript.

4-2. The mouse model is not adequately described. When was the DHEA injection (s) given? It was not explained why hCG was given before mating (lots of background, poor quality samples)

Response: We thank the reviewer for this important comment. In response, we have clarified and expanded the description of the mouse model in the **Methods** section in the revised manuscript (revised text in bold):

“C57BL/6J **female** mice at 3 weeks of age were obtained from Beijing Vital River Laboratory Animal Technology (Beijing, China). All mice always had access to food, water, and were raised in a controlled environment with a 12-hour light/dark cycle, at a room temperature of 20°C-25°C, and with a humidity of 55% ± 10%. **According to previous literature reports⁵², 3-week-old C57BL/6J female mice were given a daily subcutaneous injection of DHEA (6 mg/100 g body weight; D4000-10 g, Sigma Aldrich, USA) dissolved in 0.1 mL of sesame oil for 21 days to establish the PCOS mouse model, whose body weights were recorded every other day during treatment. From day 11 of injection, estrous cycles were monitored daily by vaginal cytology. At the study endpoint, glucose tolerance test (GTT) and insulin tolerance test (ITT) were performed, followed by orbital blood collection for serum hormone measurements (testosterone [T], estradiol [E2], and luteinizing hormone [LH]). Ovaries were processed for histopathological examination with hematoxylin and eosin (H&E) staining to evaluate cystic follicle formation and corpora lutea counts for PCOS-like phenotype validation.**”

After model establishment, DHEA-treated female mice were paired continuously (1:1) with proven fertile males. No vaginal plugs were detected in the DHEA group after 7 days of cohabitation. Subsequent controlled mating trials with fertile males demonstrated that DHEA-treated mice exhibited prolonged time-to-conception and reduced litter sizes (**Reviewer Figure 9**), which impairs normal mating and limits our ability to obtain sufficient biological replicates for downstream analysis. To standardize reproductive evaluation and ensure model validity, we implemented controlled ovarian

stimulation using low-dose gonadotropins (5 IU PMSG followed by 5 IU hCG 46-48 hours later) to induce synchronized estrus before timed mating with fertile males. To ensure data reliability, we rigorously controlled for the variable hCG, maintaining consistent experimental conditions between the control and treatment groups. We also expanded the sample size to strengthen the statistical power and credibility of our findings.

Reviewer Figure 9. Birth time and the number of pups in the first litters after mating ($n = 7$).

4-3. *The estrous cycles seem odd in DHEA mice, usually the mice are in diestrus*
doi.org/10.3389/fendo.2022.1030151

Response: As described in our response to **Comments 4-2**, 3-week-old C57BL/6J female mice were given a daily subcutaneous injection of DHEA (6 mg/100 g body weight; D4000-10 g, Sigma Aldrich, USA) dissolved in 0.1 mL of sesame oil for 21 days to establish the PCOS mouse model. The estrous cycle of DHEA mice was disrupted and maintained in estrus, which is consistent with published studies from us³³ and other research groups³⁴⁻³⁹ using the same treatment for 3-week-old C57BL/6J female mice. For the PCOS mouse model induced by subcutaneous injection of DHEA for 60-90 days in 3-week-old C57BL/6J female mice, the estrous cycle can be maintained in proestrus, estrus, diestrus, or metestrus for a long time⁴⁰, suggesting that the disappearance of cycle change is an important manifestation of disrupted estrous cycle. In the study mentioned by the reviewer⁴¹, they applied daily subcutaneous injection of DHEA in 6-week-old C57BL/6J female mice for 35 days (5 weeks). The different ages of mice and treatment time may be the possible reasons for the

differences in the estrous cycle between those studies.

4-4. Some of the results are not clear (ratios etc)

Response: We have thoroughly reviewed the manuscript, revised certain figures, and refined the presentation of insufficiently detailed results. For instance, scale bars are added to all immunostaining results (e.g., **Fig. 1a-b**). WB indicates each biological replicate clearly and displays the molecular weight markers corresponding to the target protein (e.g., **Fig. 3a-b**). Detailed dosage information for all administered medications is provided (e.g., **Fig. 5b-c**). The title, axis labels, and figure legends have been rephrased thoroughly to ensure self-explanatory.

Fig. 1a-b:

a, Representative images showing Ki67 immunohistochemistry (IHC) staining in the human mid-secretory endometrium. *b*, Immunofluorescence (IF) staining of cell morphologies in human endometrial stromal cells during induced decidualization. Decidualization was induced by the treatment of 0.5 μ M cAMP and 1 μ M MPA. The F-actin cytoskeleton was visualized by rhodamine phalloidin staining. Scale bar: 100 μ m.

Fig. 3a-b:

a-b, Protein levels of ER α and H3K181a in Ishikawa cells ($n = 6$ for each group).

Fig. 5b-c:

b, Implantation sites visualized by blue dye in naturally conceived mice with different Oxamate (0.3 - 3 mg/kg) treatments. Black arrows indicate the implantation sites. *c*, The number of implantation sites in Oil and DHEA mice with different Oxamate treatments on day 5.

4-5. The testosterone levels are also surprizingly high in PCOS and control mice.

Response: We sincerely apologize for mistakenly reporting testosterone levels. We have carefully corrected all affected values in the revised manuscript and verified them against our original records (**Supplementary Fig. 6b**). These values are also comparable to previously published results^{36,42-44}.

b, Serum testosterone (T) levels in Oil and DHEA groups ($n = 5$ for Oil group and $n = 6$ for DHEA group).

4-6. In many eperiment only stement at least $N=3$ is stated. More accuracy n is needed.

Response: We thank the reviewer for this comment and provided the exact n of biological replicates in each group in all applicable figures in the revised manuscript.

We avoid including imprecise statements such as “at least $n = 3$ ”.

For instance:

Fig. 1c, Representative images illustrating protein levels of IGFBP1 in human proliferative endometrial stromal cells during induced decidualization ($n = 5$ for each group).

Fig. 2b, Intracellular lactate levels in the human mid-secretory endometrium ($n = 17$ for the control group, and $n = 12$ for the PCOS group).

References

1. Galle, E., *et al.* H3K18 lactylation marks tissue-specific active enhancers. *Genome biology* **23**, 207 (2022).
2. Hu, X., *et al.* Dux activates metabolism-lactylation-MET network during early iPSC reprogramming with Brg1 as the histone lactylation reader. *Nucleic acids research* **52**, 5529-5548 (2024).
3. Tran, D.N., *et al.* GRB2 regulation of essential signaling pathways in the endometrium is critical for implantation and decidualization. *Nat Commun* **16**, 2192 (2025).
4. Kim, T.H., *et al.* Loss of HDAC3 results in nonreceptive endometrium and female infertility. *Sci Transl Med* **11**(2019).
5. Qiao, J., *et al.* Histone H3K18 and Ezrin Lactylation promote renal dysfunction in sepsis-associated acute kidney injury. *Advanced Science* **11**, 2307216 (2024).
6. Li, L., *et al.* Glis1 facilitates induction of pluripotency via an epigenome–metabolome–epigenome signalling cascade. *Nature metabolism* **2**, 882-892 (2020).
7. Xing, G., *et al.* MAP2K6 remodels chromatin and facilitates reprogramming by activating Gatad2b-phosphorylation dependent heterochromatin loosening. *Cell Death & Differentiation* **29**, 1042-1054 (2022).
8. Wang, N., *et al.* Histone Lactylation Boosts Reparative Gene Activation Post-Myocardial Infarction. *Circ Res* **131**, 893-908 (2022).
9. Rho, H., Terry, A.R., Chronis, C. & Hay, N. Hexokinase 2-mediated gene expression via histone lactylation is required for hepatic stellate cell activation and liver fibrosis. *Cell Metab* **35**, 1406-1423.e1408 (2023).
10. Xu, B., *et al.* Aquaporin 9: Exacerbation of Vulnerable Carotid Plaque Formation. *Biotechnology and Applied Biochemistry* (2025).
11. Qi, Y., *et al.* Indole-3-carbinol stabilizes p53 to induce miR-34a, which targets LDHA to block aerobic glycolysis in liver cancer cells. *Pharmaceuticals* **15**, 1257 (2022).
12. Zhou, X., *et al.* Icaritin activates p53 and inhibits aerobic glycolysis in liver

- cancer cells. *Chemico-Biological Interactions* **392**, 110926 (2024).
13. Bustin, S.A., *et al.* The MIQE Guidelines: Minimum Information for Publication of Quantitative Real-Time PCR Experiments. (Oxford University Press, 2009).
 14. Ho, K.H. & Patrizi, A. Assessment of common housekeeping genes as reference for gene expression studies using RT-qPCR in mouse choroid plexus. *Scientific reports* **11**, 3278 (2021).
 15. Ayakannu, T., *et al.* Validation of endogenous control reference genes for normalizing gene expression studies in endometrial carcinoma. *MHR: Basic science of reproductive medicine* **21**, 723-735 (2015).
 16. Kaya-Okur, H.S., *et al.* CUT&Tag for efficient epigenomic profiling of small samples and single cells. *Nat Commun* **10**, 1930 (2019).
 17. Li, F., *et al.* Positive feedback regulation between glycolysis and histone lactylation drives oncogenesis in pancreatic ductal adenocarcinoma. *Molecular Cancer* **23**, 90 (2024).
 18. Wu, Y., Wang, Y., Dong, Y., Sun, L.V. & Zheng, Y. Lactate promotes H3K18 lactylation in human neuroectoderm differentiation. *Cellular and Molecular Life Sciences* **81**, 459 (2024).
 19. Zhao, Y., *et al.* Lactate modulates zygotic genome activation through H3K18 lactylation rather than H3K27 acetylation. *Cellular and Molecular Life Sciences* **81**, 298 (2024).
 20. Li, F., *et al.* Positive feedback regulation between glycolysis and histone lactylation drives oncogenesis in pancreatic ductal adenocarcinoma. *Mol Cancer* **23**, 90 (2024).
 21. Babayev, S.N., *et al.* Thrombin alters human endometrial stromal cell differentiation during decidualization. *Reproductive Sciences* **26**, 278-288 (2019).
 22. Mei, J., *et al.* CXCL16/CXCR6 interaction promotes endometrial decidualization via the PI3K/AKT pathway. *Reproduction* **157**, 273-282 (2019).
 23. Gibson, D.A., Simitsidellis, I., Kelepouri, O., Critchley, H.O. & Saunders, P.T.

- Dehydroepiandrosterone enhances decidualization in women of advanced reproductive age. *Fertility and sterility* **109**, 728-734. e722 (2018).
24. Li, R., *et al.* Prevalence of polycystic ovary syndrome in women in China: a large community-based study. *Human reproduction* **28**, 2562-2569 (2013).
 25. Sermondade, N., *et al.* Female obesity is negatively associated with live birth rate following IVF: a systematic review and meta-analysis. *Human reproduction update* **25**, 439-451 (2019).
 26. Xu, X., *et al.* Endometrial stromal Menin supports endometrial receptivity by maintaining homeostasis of WNT signaling pathway through H3K4me3 during WOI. *Commun Biol* **8**, 995 (2025).
 27. Yu, H.F., *et al.* Ptn functions downstream of C/EBP β to mediate the effects of cAMP on uterine stromal cell differentiation through targeting Hand2 in response to progesterone. *J Cell Physiol* **233**, 1612-1626 (2018).
 28. Li, Q., *et al.* The antiproliferative action of progesterone in uterine epithelium is mediated by Hand2. *Science* **331**, 912-916 (2011).
 29. Xin, Q., *et al.* Polycomb subunit BMI1 determines uterine progesterone responsiveness essential for normal embryo implantation. *The Journal of Clinical Investigation* **128**, 175-189 (2018).
 30. Cheng, J.-G., Chen, J.R., Hernandez, L., Alvord, W.G. & Stewart, C.L. Dual control of LIF expression and LIF receptor function regulate Stat3 activation at the onset of uterine receptivity and embryo implantation. *Proceedings of the National Academy of Sciences* **98**, 8680-8685 (2001).
 31. Okada, H., Tsuzuki, T. & Murata, H. Decidualization of the human endometrium. *Reproductive medicine and biology* **17**, 220-227 (2018).
 32. Ramathal, C.Y., Bagchi, I.C., Taylor, R.N. & Bagchi, M.K. Endometrial decidualization: of mice and men. in *Seminars in reproductive medicine*, Vol. 28 017-026 (© Thieme Medical Publishers, 2010).
 33. Qi, X., *et al.* Gut microbiota–bile acid–interleukin-22 axis orchestrates polycystic ovary syndrome. *Nature medicine* **25**, 1225-1233 (2019).
 34. Ni, F., *et al.* Proteome-wide Mendelian randomization and functional studies

- uncover therapeutic targets for polycystic ovarian syndrome. *The American Journal of Human Genetics* **111**, 2799-2813 (2024).
35. Shen, H., Xu, X., Fu, Z., Xu, C. & Wang, Y. The interactions of CAP and LYN with the insulin signaling transducer CBL play an important role in polycystic ovary syndrome. *Metabolism* **131**, 155164 (2022).
 36. Guo, Z., Chen, X., Feng, P. & Yu, Q. Short-term rapamycin administration elevated testosterone levels and exacerbated reproductive disorder in dehydroepiandrosterone-induced polycystic ovary syndrome mice. *J Ovarian Res* **14**, 64 (2021).
 37. Emidio, G.D., *et al.* Methylglyoxal-Dependent Glycative Stress and Deregulation of SIRT1 Functional Network in the Ovary of PCOS Mice. *Cells* **9**(2020).
 38. Dou, L., *et al.* The effect of cinnamon on polycystic ovary syndrome in a mouse model. *Reprod Biol Endocrinol* **16**, 99 (2018).
 39. Yan, S., *et al.* Nanocomposites based on nanoceria regulate the immune microenvironment for the treatment of polycystic ovary syndrome. *J Nanobiotechnology* **21**, 412 (2023).
 40. Liu, Y., *et al.* Artemisinins ameliorate polycystic ovarian syndrome by mediating LONP1-CYP11A1 interaction. *Science* **384**, eadk5382 (2024).
 41. Wang, X., *et al.* Effects of dehydroepiandrosterone alone or in combination with a high-fat diet and antibiotic cocktail on the heterogeneous phenotypes of PCOS mouse models by regulating gut microbiota. *Frontiers in endocrinology* **13**, 1030151 (2022).
 42. Shi, X.J., *et al.* Treatment of polycystic ovary syndrome and its associated psychiatric symptoms with the Mongolian medicine Nuangong Qiwei Pill and macelignan. *J Ethnopharmacol* **317**, 116812 (2023).
 43. Gu, M., *et al.* Dingkun pill alleviates metabolic abnormalities in polycystic ovary syndrome through brown adipose tissue activation. *J Ovarian Res* **16**, 176 (2023).
 44. Qi, X., Yun, C., Liao, B., Qiao, J. & Pang, Y. The therapeutic effect of

interleukin-22 in high androgen-induced polycystic ovary syndrome. *J Endocrinol* **245**, 281-289 (2020).

We want to thank the reviewers sincerely for their constructive comments and criticism that helped us substantially improve the quality and impact of our manuscript. We have thoroughly revised the manuscript to address all the reviewers' concerns. Our main addition includes:

1. Evaluate the impact of BMI on our observations extensively and confirm that the excessive endometrial ER α and H3K181a are PCOS-dependent instead of BMI-dependent.
2. Increase the clarity of the manuscript by adding a detailed description of the organoid generation and propensity score matching (PSM).
3. Rephrase the **Discussion** section, highlighting recent publications indicating endometrial dysfunction as an independent contributor of ovaries to PCOS infertility, and link the therapeutic effect of metformin on PCOS to our findings.

Together with the revised manuscript, we submit below a point-by-point response detailing actions taken to address each comment. We hope that you will find the revision thorough and satisfactory. We welcome any additional questions.

With our sincere appreciation,

The authors.

Please see our detailed responses below in blue.

REVIEWER COMMENTS

Reviewer #1 (Remarks to the Author):

Dear Authors,

I would like to thank the authors for their detailed and thoughtful responses to the comments raised during the review process. I have carefully evaluated the revised manuscript and the point-by-point replies. The authors have satisfactorily addressed all major concerns.

Specifically, the revised introduction now clearly articulates the research hypothesis and objectives. The authors have appropriately increased the number of biological replicates and explained the rationale for variation in sample size across experiments. Methodological rigour has been strengthened, including validation of primary cell viability, lactate quantification assays, qRT-PCR compliance with MIQE guidelines, and improved documentation for immunostaining and Western blot analyses.

Concerns regarding reference protein variability, CUT&Tag replicates, and the interpretation of immunohistochemistry have been resolved through repeat experiments and improved clarity in figure legends and supplementary data.

Regarding the authors' query on including multiple normalisation figures for qRT-PCR: In my opinion, presenting one well-validated reference gene per species in the main figures is sufficient. The additional normalisation results included in the supplementary materials provide adequate transparency for readers.

In summary, I find the revised manuscript to be significantly improved. The authors have demonstrated methodological care, appropriate statistical reasoning, and transparency in reporting. I recommend acceptance of the manuscript.

Response: We thank the reviewer for the kind comments, and we are glad that the reviewer agrees that the manuscript has been substantially improved following the reviewers' constructive comments. We are also grateful for the recommendation of acceptance. Based on the reviewer's suggestion, we have retained the human qRT-PCR results, normalized to *GAPDH*, and the murine qRT-PCR results, normalized to *Tbp*, in the main figures. Additionally, the human qRT-PCR results, normalized to *ACTB*, and the murine qRT-PCR results, normalized to *Gapdh*, were included as **Supplementary Data Figure 5**.

Reviewer #2 (Remarks to the Author):

The authors have adequately addressed my previous comments, and I find that the manuscript has improved significantly as a result.

Response: We thank the reviewer for recognizing the improvement of our revised manuscript, and we are glad that the reviewer feels we have adequately addressed all the raised comments.

Reviewer #3 (Remarks to the Author):

Response: We thank the reviewer for co-reviewing our work.

Reviewer #4 (Remarks to the Author):

The paper has improved significantly and the authors have rigorously replied to my concerns and comments. The data provides novel approach to improve endometrial receptivity in women with polycystic ovary syndrome (PCOS) by targeting Era/histone lactylation. I congratulate the team to be able to generate also new data for this manuscript.

Response: We thank the reviewer for recognizing our efforts to improve the manuscript, and we believe that the manuscript has benefited substantially from the reviewers' constructive comments.

Some additional comments:

1. What is the role of BMI? For example the BMIs for the patients donating their endometrium for the in vitro studies differ from the controls. Recent studies imply that BMI also has an effect on the endometrium and epithelial cell driven organoids and may explain some changes between the study groups.

<https://pubmed.ncbi.nlm.nih.gov/40499150/>

Even though the BMIs were inside the normal BMI category the asian ethnicity experience insulin resistance with lower BMI that is most likely the case with the samples also used here.

Could the authors strengthen their results to show PCOS-independent (and not BMI dependent) difference between cases and controls. For example show if the in vitro samples were matched for BMI.

Response: We thank the reviewer for this important suggestion, and we fully agree that BMI is a critical feature to be considered when we compare human endometrial samples between PCOS and controls. As suggested by the reviewer, PSM (1:1 nearest neighbor) was performed with a caliper width set to 0.25 times the standard deviation of BMI (0.8), following Rosenbaum and Rubin's recommended approach to minimize bias for in vitro samples (**Supplementary Data Figure 9**). The elevated gene expression levels of *ESR1*, *MSX1*, *MSX2*, and *LTF* were retained post-matching (**Supplementary Data Figure 9a-b**). No significant differences were observed in gene expression levels of *PGR*, *AREG*, and *IHH*, which were also consistent before and after PSM (**Supplementary Data Figure 9a-b**). ER α and H3K18la levels were also confirmed to be increased in PCOS after PSM (**Supplementary Data Figure 9c-e**). All these observations indicated that increased human endometrial ER α and H3K18la levels were PCOS-dependent instead of BMI-dependent.

a

Ages and BMIs of the two groups before and after matching
(donors of endometrial samples used for qRT-PCR)

Item	Before PSM			After PSM		
	Control group	PCOS group	P value	Control group	PCOS group	P value
	(n=25)	(n=12)		(n=11)	(n=11)	
Age of female (year)	32.5±3.1	31.6±3.7	0.396	32.5±3.1	32.1±3.2	0.752
BMI (kg/m ²)	21.1±3.0	24.4±3.3	0.004	23.9±2.3	24.9±2.1	0.083

b**c**

Ages and BMIs of the two groups before and after matching
(donors of endometrial samples used for WB)

Item	Before PSM			After PSM		
	Control group	PCOS group	P value	Control group	PCOS group	P value
	(n=25)	(n=12)		(n=6)	(n=6)	
Age of female (year)	32.5±3.1	31.6±3.7	0.396	32.2±2.8	32.0±3.5	0.912
BMI (kg/m ²)	21.1±3.0	24.4±3.3	0.004	23.8±2.5	24.1±2.3	0.831

d**e**
Supplementary Data Figure 9. Elevated endometrial ER α and H3K181a levels are PCOS-dependent but not BMI-dependent. **a**, Ages and BMIs of the two groups before and after matching (donors of endometrial samples used for qRT-PCR). **b**, Relative mRNA levels of *ESR1*, *MSX1*, *MSX2*, *LTF*, *PGR*, *AREG*, and *IHH* in the human mid-secretory endometrium after PSM ($n = 11$). **c**, Ages and BMIs of the two groups before and after matching (donors of endometrial samples used for WB). **d-e**, Protein levels of ER α and H3K181a in the human mid-secretory endometrium after PSM ($n = 6$). PSM was performed using BMI as the matching variable.

Additionally, we have expanded our cohort by recruiting additional PCOS patients and stratified subjects based on BMI: a normal-weight group (BMI < 24 kg/m², N-PCOS) and an obese group (BMI \geq 24 kg/m², O-PCOS) with matched controls. We have carefully examined those critical features mentioned in our manuscript, including expression of genes relevant to estrogen and progesterone signaling as well as H3K181a levels. The upregulation of ER α , estrogen-responsive genes, and H3K181a was consistently observed in N-PCOS (versus N-control) and O-PCOS (versus O-control), further confirming that our findings were PCOS-dependent but not BMI-dependent (**Supplementary Data Figure 10**).

Supplementary Data Figure 10. Endometrial ER α and H3K181a levels are consistently upregulated in PCOS compared to controls with matched BMIs. **a**, BMIs of donors and relative mRNA levels of *ESR1*, *MSX1*, *MSX2*, *LTF*, *PGR*, *AREG*, and *IHH* in the human mid-secretory endometrium of normal-weight controls (N-Control) and PCOS (N-PCOS) ($n = 16$ for N-Control and $n = 8$ for N-PCOS). **b-d**, BMIs of donors and protein levels of ER α and H3K181a in the human mid-secretory endometrium of N-Control and N-PCOS ($n = 6$). **e**, BMIs of donors and relative mRNA levels of *ESR1*, *MSX1*, *MSX2*, *LTF*, *PGR*, *AREG*, and *IHH* in the human mid-secretory endometrium of obese controls (O-Control) and PCOS (O-PCOS) ($n = 16$ for O-Control and $n = 8$ for O-PCOS). **f-h**, BMIs of donors and protein levels of ER α and H3K181a in the human mid-secretory endometrium of O-Control and O-PCOS ($n = 6$).

Lastly, we believe that the discussion on BMI represents a great supplement to our manuscript and will further enhance the clarity and reliability, which has been added to the **Discussion** section in the revised manuscript:

“When comparing endometrial samples between women with PCOS and controls, it is necessary to consider BMI. Even though the BMIs of PCOS donors for our in vitro studies were within the normal BMI category, the Asian ethnicity experiences insulin resistance with lower BMI, which might obscure the conclusion⁶². To confirm our findings to be PCOS-dependent instead of BMI-dependent, PSM was further applied to sample donors of the PCOS and control groups to make BMIs of the two groups comparable. The excessive ER α and H3K181a were still observed in PCOS after PSM (Supplementary Data). We also collected additional samples, confirming that ER α and H3K181a are always higher in PCOS compared to controls within different BMI categories (Supplementary Data).”

Please give also out the protocol for the organoid generation.

Response: We thank the reviewer for this suggestion, which helps us to improve the clarity of our manuscript. We have added the protocol for the organoid generation into the **Methods** section:

“Endometrial tissues were minced into small fragments under sterile conditions and subjected to enzymatic digestion in 5 ml PBS containing 1 mg/ml collagenase IV (Sigma-Aldrich, USA, C4-28) at 37 °C for 20-30 minutes with gentle agitation. Digestion was terminated by adding an equal volume of advanced DMEM/F-12 (Gibco, USA, 12634028). The suspension was gently pipetted and sequentially filtered through 100 μ m (Falcon, USA, 352360) and 40 μ m (Falcon, USA, 352340) cell strainers. Glandular fragments were collected via backwashing of the 40 μ m strainer, resuspended in ice-cold Matrigel (Corning, USA, 536231), and plated into 48-well plates. After polymerization at 37 °C, the Matrigel domes were overlaid with 250 μ l of organoid expansion medium. The medium consisted of advanced DMEM/F-12 supplemented with 1% N2 (ThermoFisher, USA, 17502048), 1% B27 (ThermoFisher, USA, 17504044), 1% Glutamax (Gibco, USA, 35050061), 500 ng/ml R-Spondin 1 (Novoprotein, China, CX83), 100 ng/ml Noggin (Novoprotein, China, CB89), 1.25 mM N-Acetylcysteine (Sigma-Aldrich, USA, A0737), 10 mM Nicotinamide (Sigma-Aldrich, USA, N0636), 25 ng/ml HGF (Novoprotein, China, CJ72), 50 ng/ml EGF (Novoprotein, China, C029), 100 ng/ml FGF10 (Novoprotein, China, CR11), 10 μ M Y-27632 (MCE, USA, HY-10583), 500 nM A83-01 (MCE, USA, HY-

10432), 1× ITS (ThermoFisher, USA, 41400045), and 1% Penicillin/Streptomycin (Gibco, USA, 15140122). The medium was refreshed every 2-3 days. Organoids were passaged every 7-10 days by mechanical disruption and re-embedded in Matrigel. For subsequent analyses, organoids were harvested using Cell Recovery Solution (Corning, USA, 354253).”

2. Some recent key papers from the field could be cited as they support the endometrial dysfunction in PCOS independently from the ovaries and that epithelium could be main target given the higher proportion of epithelial cells in PCOS endometrium. Also the role of Metformin has also been investigated to “rescue” PCOS endometrium. Can the authors elucidate how metformin treatment would fit in their theory.

<https://pmc.ncbi.nlm.nih.gov/articles/PMC12176659/>

<https://pubmed.ncbi.nlm.nih.gov/40344073/>

Response: We thank the reviewer for this suggestion. We agree that including recent findings that support the endometrial dysfunction in PCOS as an independent feature of the ovaries will significantly strengthen the discussion of our manuscript. These key papers are now highlighted and cited by the revised manuscript in the **Discussion** section (revised text in bold):

“PCOS is a common endocrine disorder, and growing evidence suggests that PCOS negatively impacts fertility and pregnancy outcomes⁵. Besides defective qualities of oocytes and embryos, endometrial dysfunction may also contribute to PCOS infertility^{3,6,42,43}. **For instance, the endometrium of women with PCOS has been reported to have a higher proportion of epithelial cells, indicating the epithelium as a main target of PCOS-related abnormalities, which might adversely affect implantation⁴².**”

We hypothesize that metformin could affect the endometrial function in PCOS via targeting ER α and lactate at the same time. However, the underlying mechanisms seem to be controversial since metformin may reduce ER α expression but also induce lactate accumulation. Therefore, further experimental efforts are needed to illustrate the linkage between metformin treatment and changes in endometrial ER α /H3K18la. This discussion has been added to the **Discussion** section in the revised manuscript:

“Recent studies highlight the impact of metformin in restoring endometrial health in patients with PCOS^{42,53}. Along with our findings, metformin may rescue endometrial function in PCOS

via regulating ER α . It has been reported that metformin can reduce ER α expression in the endometrium of diabetic women with endometrial cancer and Ishikawa cells^{54,55}. However, as a classic therapeutic drug for diabetes, metformin inhibits mitochondrial oxidative phosphorylation, potentially causing lactate accumulation⁵⁶⁻⁵⁹. In addition, metformin ameliorates liver fibrosis in mice via the enrichment of *Lactobacillus* sp. MF-1 in the gut microbiota⁶⁰, and emerging evidence suggests that the gut microbiota and derived metabolism may influence the microenvironment of the female reproductive tract⁶¹, which could potentially impact endometrial lactate levels. Thus, the interplay between ER α and K1a is complex during the pathogenesis and treatment of PCOS. Extra research efforts are needed to understand how metformin improves endometrial receptivity via regulating ER α /K1a in PCOS.”

3. *Supplementary table 1. Please explain PSM for the readers.*

Response: We thank the reviewer for this suggestion. We have added the explanation of PSM into the **Methods** section and the legend of **Supplementary Table 1** (revised text in bold):

Methods section

“To address potential confounders, we performed PSM using covariates **that have been established to be associated** with reproductive outcomes in PCOS and infertility research. These included female age, BMI, infertility duration, miscarriage history, and male age, all of which are associated with pregnancy success or live birth rates⁶³⁻⁶⁷.”

Supplementary Table 1

“Propensity score matching (PSM) was performed using female age, BMI, infertility duration, miscarriage history, and male age as matching variables.”

Reference

1. Luyckx, L., *et al.* Prenatally androgenized PCOS mice have ovary-independent uterine dysfunction and placental inflammation aggravated by high-fat diet. *Sci Adv* **11**, eadu3699 (2025).
2. Eriksson, G., *et al.* Single-cell profiling of the human endometrium in polycystic ovary syndrome. *Nat Med* **31**, 1925-1938 (2025).

3. Markowska, A., *et al.* Does Metformin affect ER, PR, IGF-1R, β -catenin and PAX-2 expression in women with diabetes mellitus and endometrial cancer? *Diabetol Metab Syndr* **5**, 76 (2013).
4. Zhang, J., *et al.* Role of metformin in inhibiting estrogen-induced proliferation and regulating ER α and ER β expression in human endometrial cancer cells. *Oncol Lett* **14**, 4949-4956 (2017).
5. Boucaud-Maitre, D., *et al.* Lactic acidosis: relationship between metformin levels, lactate concentration and mortality. *Diabet Med* **33**, 1536-1543 (2016).
6. See, K.C. Metformin-associated lactic acidosis: A mini review of pathophysiology, diagnosis and management in critically ill patients. *World J Diabetes* **15**, 1178-1186 (2024).
7. Chang, M.Y., *et al.* Metformin induces lactate accumulation and accelerates renal cyst progression in Pkd1-deficient mice. *Hum Mol Genet* **31**, 1560-1573 (2022).
8. Fadden, E.J., Longley, C. & Mahambrey, T. Metformin-associated lactic acidosis. *BMJ Case Rep* **14**(2021).
9. Yang, T., Guan, Q., Shi, J.S., Xu, Z.H. & Geng, Y. Metformin alleviates liver fibrosis in mice by enriching *Lactobacillus* sp. MF-1 in the gut microbiota. *Biochim Biophys Acta Mol Basis Dis* **1869**, 166664 (2023).